# Oxytocin functions as a spatiotemporal filter for excitatory synaptic inputs to VTA dopamine neurons

Lei Xiao, Michael F Priest, Yevgenia Kozorovitskiy*

Department of Neurobiology, Northwestern University, Evanston, United States

**Abstract** The experience of rewarding or aversive stimuli is encoded by distinct afferents to dopamine (DA) neurons of the ventral tegmental area (VTA). Several neuromodulatory systems including oxytocin regulate DA neuron excitability and synaptic transmission that process socially meaningful stimuli. We and others have recently characterized oxytocinergic modulation of activity in mouse VTA DA neurons, but the mechanisms underlying oxytocinergic modulation of synaptic transmission in DA neurons remain poorly understood. Here, we find that oxytocin application or optogenetic release decrease excitatory synaptic transmission, via long lasting, presynaptic, endocannabinoid-dependent mechanisms. Oxytocin modulation of excitatory transmission alters the magnitude of short and long-term depression. We find that only some glutamatergic projections to DA neurons express CB1 receptors. Optogenetic stimulation of three major VTA inputs demonstrates that oxytocin modulation is limited to projections that show evidence of CB1R transcripts. Thus, oxytocin gates information flow into reward circuits in a temporally selective and pathway-specific manner.

DOI: https://doi.org/10.7554/eLife.33892.001

## Introduction

Dopamine (DA) neurons in the ventral tegmental area (VTA) play a pivotal role in signaling reward-related stimuli (*Wise, 2004*). Alterations in the function of VTA DA neurons have been linked to drug abuse (*Kauer and Malenka, 2007*; *Niehaus et al., 2009*), as well as neurodevelopmental and psychiatric disorders including autism spectrum disorders (ASDs) (*Bariselli et al., 2016*), schizophrenia, and depression (*Grace, 2016*). DA neurons exhibit two activity patterns in vivo – low frequency tonic firing (1–5 Hz) and burst, or phasic, firing patterns (*Floresco et al., 2003*; *Grace and Bunney, 1984*). Phasic activity of DA neurons is observed when an animal is in an environment with strong reward salience (*Schultz, 1998*; *Schultz et al., 2015*), resulting in large transient increases in DA concentration in the ventral striatum. These DA increases have been directly assayed using fast-scan cyclic voltammetry (*Tsai et al., 2009*) and are well-matched by recent optical imaging studies that demonstrate unpredicted reward-associated calcium transients in the axons of VTA DA neurons expressing genetically encoded calcium sensors (*Howe and Dombeck, 2016*). Glutamatergic afferents to DA neurons are thought to control transitions between tonic and phasic activity (*Canavier and Landry, 2006*; *Floresco et al., 2003*; *Overton and Clark, 1997*). For the VTA, these inputs arise from multiple brain regions, including the medial prefrontal cortex (mPFC), lateral habenular nucleus (LHb) and pedunculopontine tegmental nucleus (PPN) (*Geisler et al., 2007*; *Good and Lupica, 2009*; *Morales and Margolis, 2017*).

Neuronal activity and glutamatergic synaptic transmission in the VTA are tightly regulated by a diverse group of neuromodulators acting on G protein-coupled receptors (GPCRs). VTA DA neurons express numerous GPCRs, including dopamine Drd2 receptor (*Vallone et al., 2000*), as well as receptors for serotonin (*Doherty and Pickel, 2000*), corticotropin-releasing factor (*Ungless et al.,*

**\*For correspondence:**
Yevgenia.Kozorovitskiy@
northwestern.edu

**Competing interests:** The authors declare that no competing interests exist.

**eLife digest** The mammalian brain contains millions of nerve cells or neurons that communicate with each other via a process called neurotransmission. To send a message to its neighbor, a neuron releases a chemical called a neurotransmitter into the space between the cells. The neurotransmitter then binds to receiver proteins on the target cell. Another group of chemicals, known as neuromodulators, regulate this process, adjusting the way that neurons respond to neurotransmitters. In doing so, they help regulate many types of behavior in mammals.

The neuromodulator oxytocin, for example, has earned the nickname 'the love hormone' because it promotes social behavior and bonding. It does this in part by altering the activity of neurons in a brain region called the ventral tegmental area (VTA). These neurons produce the brain's main reward signal, dopamine, which is itself a neuromodulator. But exactly how oxytocin affects the activity of dopamine-producing neurons is unclear.

By recording from individual neurons in slices of mouse brain tissue, Xiao et al. show that oxytocin filters inputs to dopamine neurons in the VTA. It does this by making the dopamine neurons release another group of reward signals, known as endocannabinoids. These are the brain's own version of the chemicals found inside cannabis plants. The endocannabinoids bind to neurons that provide input to the VTA dopamine neurons. Some of these input neurons normally activate the VTA by releasing a neurotransmitter called glutamate. However, the binding of endocannabinoids decreases their ability to do this, and thereby lowers the activation of the VTA dopamine neurons.

But not all glutamate neurons are sensitive to endocannabinoids. Moreover, oxytocin affects glutamate neurons that fire repeatedly less than it affects those that fire only occasionally. Oxytocin thus acts as a filter. It allows certain inputs – those that are repeatedly active and those that are insensitive to endocannabinoids – to continue activating VTA dopamine neurons. At the same time, it weakens the influence of other inputs. Dopamine release in the VTA drives drug abuse and addiction. Understanding how oxytocin affects VTA neurons may thus open up new avenues for the treatment of addiction disorders.

DOI: https://doi.org/10.7554/eLife.33892.002

2003), orexin (*Korotkova et al., 2003*), vasopressin, and oxytocin (*Skuse and Gallagher, 2009*; *Xiao et al., 2017*), among others. Several GPCR families are known to modulate neuronal activity and synaptic transmission in VTA DA neurons (*Bonci and Malenka, 1999*; *Borgland et al., 2006*; *Ungless et al., 2003*), but our understanding of how neurohypophyseal peptides control synaptic transmission in the midbrain dopamine system is limited.

Oxytocin is one of the two major neurohypophyseal peptides that function centrally as a neuro-modulator and has been linked to the processing of socially rewarding stimuli via actions in the striatum and in the neocortex (*Choe et al., 2015*; *Dölen et al., 2013*; *Marlin et al., 2015*). In the nucleus accumbens, oxytocin weakens glutamatergic synaptic transmission through presynaptic mechanisms involving serotonergic inputs from the dorsal raphe nucleus (*Dölen et al., 2013*), and in the mPFC, oxytocin application dampens glutamatergic synaptic transmission through presynaptic endocannabinoid signaling (*Ninan, 2011*). Yet, in the auditory and piriform cortex, oxytocin application or evoked release dampen inhibitory transmission without altering excitatory input processing (*Mitre et al., 2016*). Oxytocin neurons residing in the paraventricular nucleus of the hypothalamus (PVN) directly project to the VTA and activate oxytocin receptors (OxtRs) to regulate the tonic activity of DA neurons and social behavior (*Beier et al., 2015*; *Hung et al., 2017*; *Tang et al., 2014*; *Xiao et al., 2017*). Consistent with oxytocin-driven enhancement in the activity of VTA DA neurons, oxytocin infusions into the VTA potentiate social reward (*Song et al., 2016*) and modulate reward-based behavior (*Mullis et al., 2013*). Given these observations, we reasoned that oxytocin is poised to regulate synaptic transmission in the VTA. Recently, OxtR agonist was found to regulate synaptic inputs to a specific subpopulation of VTA DA neurons that project to medial nucleus accumbens, altering the balance of excitation and inhibition through cellular mechanisms that remain incompletely understood (*Hung et al., 2017*). Here, we delve deeper into the function of oxytocin in regulating evoked excitatory postsynaptic currents (EPSCs) in genetically targeted VTA DA neurons by relying on pharmacological methods, optogenetics, retrograde tracing, quantitative triple-channel in

situ hybridization, and 2-photon imaging with single synapse 2-photon uncaging, or photorelease, of glutamate. We find that oxytocin inhibits excitatory synaptic transmission through OxtRs and retrograde endocannabinoid signaling in a cell-autonomous manner, with timing and effect magnitudes sufficient to modulate both short- and long-term plasticity in VTA DA neurons.

## Results

### Oxytocin inhibits evoked excitatory synaptic currents in VTA DA neurons via OxtR

To determine whether oxytocin regulates synaptic transmission in VTA DA neurons we carried out voltage-clamp recordings of tdTomato[+] or eYFP[+] neurons in horizontal slices prepared from P25-40 *Slc6a3*[i-Cre]; Ai14 or Ai32 reporter mice (*Figure 1A*). Many DA neurons in ventromedial VTA express oxytocin receptors (*Figure 1—figure supplement 1A–C*), and the majority of DA neurons in this region increase tonic activity in response to oxytocin application (*Tang et al., 2014*; *Xiao et al., 2017*), so we targeted these cells for recordings in this study. Excitatory postsynaptic currents (EPSCs) were evoked by electrical stimulation at a holding potential of −70 mV in the presence of 10 μM GABA(A) receptor antagonist (SR95531) at room temperature. Evoked EPSCs were largely blocked by the AMPA receptor antagonist NBQX and fully abolished following subsequent application of the NMDA receptor antagonist CPP (*Figure 1—figure supplement 1D*). Bath application of 1 μM oxytocin significantly reduced the amplitude of evoked EPSCs (77.43 ± 1.55% of baseline, p<0.001, paired *t*-test, *n* = 15 neurons from 10 mice) (*Figure 1A,B and F*), without changing the input resistance of VTA DA neurons or decay time constant of EPSCs (*Figure 1—figure supplement 1E–F*). This oxytocin-induced EPSC depression in VTA DA neurons was similar in males and females (♂, 79.12 ± 5.58% of baseline, *n* = 5 neurons; ♀, 77.36 ± 3.39% of baseline, *n* = 10 neurons. p=0.440, Mann-Whitney test), consistent with a prior report on sex invariance of oxytocinergic regulation of midbrain DA systems in structure and function (*Xiao et al., 2017*). To build a dose-response curve we also evaluated the effect of oxytocin on excitatory synaptic transmission at concentrations below 1 μM (0 nM, 10 nM, 100 nM, and 500 nM), and observed that oxytocin application in concentrations at or above 100 nM is sufficient to decrease EPSC amplitude in a dose-dependent manner (0 nM, 103.50 ± 3.05% of baseline; 10 nM, 94.74 ± 1.38% of baseline; 100 nM, 87.73 ± 0.88% of baseline; 500 nM, 83.31 ± 3.18% of baseline, p<0.01, Kruskal-Wallis test with Dunn's Multiple Comparison *post hoc* test, *n* = 5–8 neurons from 2 to 5 mice/group) (*Figure 1F*, *Figure 1—source data 1*). These results suggest that although oxytocin release or application enhance spontaneous tonic activity of VTA DA neurons (*Tang et al., 2014*; *Xiao et al., 2017*), the same peptide dampens excitatory synaptic transmission in VTA DA neurons.

In the central nervous system, oxytocin primarily binds to a single variant of the oxytocin receptor (OxtR) to regulate neuronal activity (*Gimpl and Fahrenholz, 2001*; *Stoop, 2012*). OxtR is expressed in some VTA DA neurons (*Hung et al., 2017*; *Vaccari et al., 1998*; *Xiao et al., 2017*). We next tested whether oxytocin regulates the excitatory synaptic transmission to VTA DA neurons by activating OxtR. In the presence of OxtR antagonist (10 μM L368,899) we observed that bath application of 1 μM oxytocin did not have a significant effect on the amplitude of evoked EPSCs (98.89 ± 3.39% of baseline, p=0.844, Wilcoxon matched-pairs signed rank test, *n* = 8 neurons from six mice) (*Figure 1C and G*, *Figure 1—source data 1*). OxtR antagonist application alone had no effect on the amplitude of evoked EPSCs (*Figure 1—figure supplement 1G*), which suggests that no oxytocin tone is present in the acute brain slice of the VTA. OxtRs primarily couple to Gα$_q$ protein effectors, which activate the phospholipase C (PLC) pathway to mobilize intracellular Ca$^{2+}$ store release (*Gimpl and Fahrenholz, 2001*; *Stoop, 2012*). We evaluated the contribution of the PLC pathway to the observed dampening of synaptic transmission using a selective PLC inhibitor U73122. In the presence of 10 μM U73122, oxytocin failed to decrease evoked EPSC amplitude (98.92 ± 5.33% of baseline, p=0.949, Wilcoxon matched-pairs signed rank test, *n* = 5 neurons from four mice) (*Figure 1D and G*, *Figure 1—source data 1*). Thus, oxytocin binds to Gα$_q$ protein-coupled OxtR to regulate synaptic transmission in VTA DA neurons through the canonical PLC pathway.

In addition to OxtR, oxytocin has been shown to bind structurally related vasopressin receptors (*Gimpl and Fahrenholz, 2001*; *Tribollet et al., 1988*). Indeed, vasopressin 1a receptor (V1aR) transcripts are present in the rodent VTA, including in DA neurons (*Dubois-Dauphin et al., 1996*;

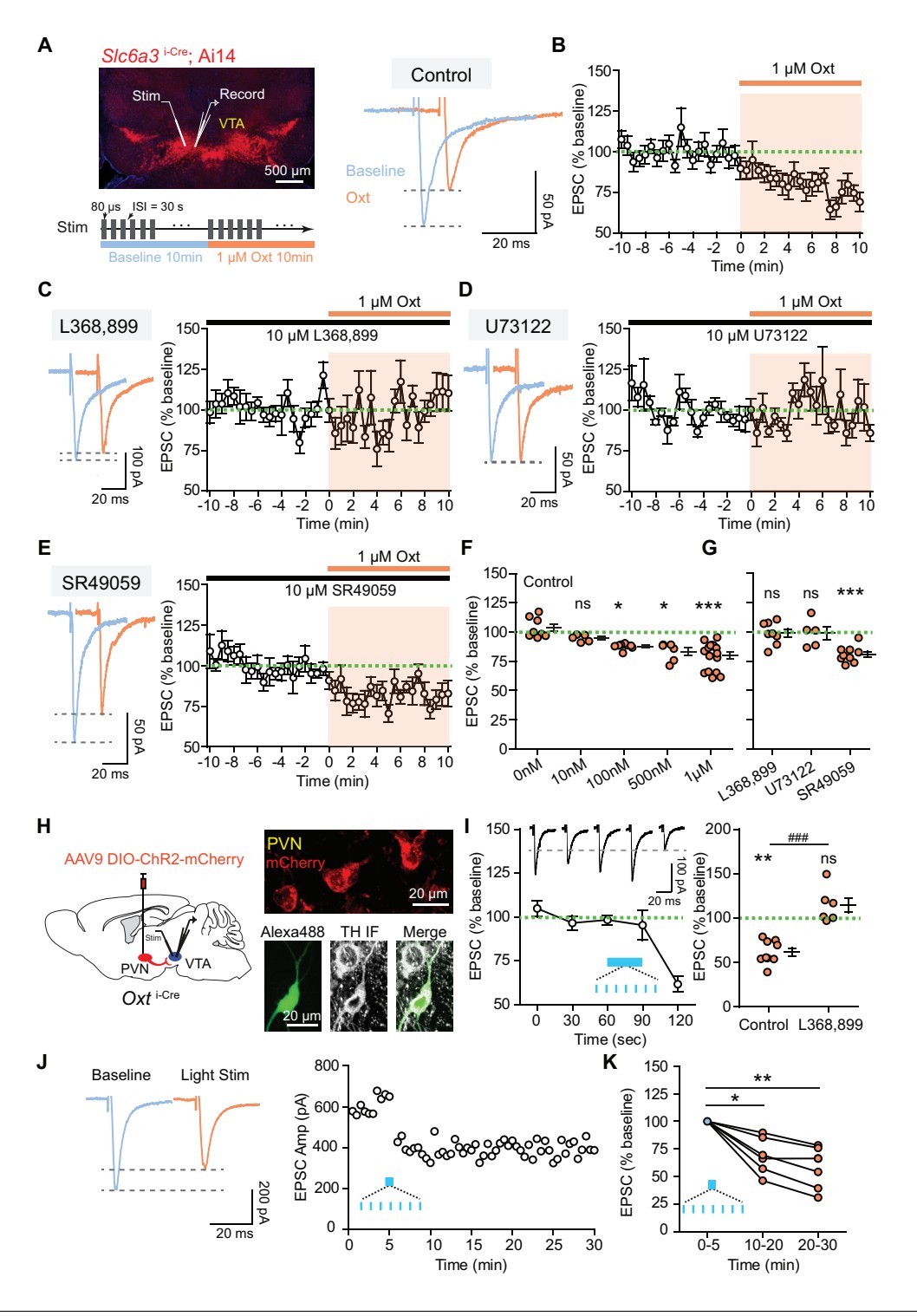

**Figure 1.** Oxytocin inhibits excitatory synaptic transmission in VTA DA neurons via the OxtR. (**A**) Left, top: an image of a horizontal brain section from a *Slc6a3*[i-Cre]; A14 mouse. A stimulation electrode (Stim) was placed ~100 μm away from the recording electrode (Record). Left, bottom: experimental protocol schematic. Right: example traces of evoked EPSCs from one VTA DA neuron during baseline and following 1 μM oxytocin application. Blue, baseline; orange, following oxytocin application. (**B**) The amplitude of evoked EPSC decreased during the application of 1 μM oxytocin. n = 15 neurons from 10 mice. Green dashed line, baseline EPSC amplitude average; shaded region, time of oxytocin application. (**C**) OxtR antagonist L368,899 blocked oxytocin-induced EPSC

*Figure 1 continued on next page*

*Figure 1 continued*

amplitude attenuation. Left: example traces from one neuron before and during oxytocin application. Right: summary data for n = 8 neurons from six mice. (D) Same as (B), but with a phospholipase C blocker U73122. n = 5 neurons from four mice. (E) Same as (B), but with a vasopressin receptor (V1aR) antagonist SR49059. n = 9 neurons from six mice. (F) Summary data for EPSC amplitude changes induced by oxytocin at different concentrations (0 nM, 10 nM, 100 nM, 500 nM, and 1 µM). For each concentration respectively, n = 8 neurons from five mice, 5 neurons from two mice, 8 neurons from four mice, 6 neurons from five mice, and 15 neurons from 10 mice. *p<0.05, ***p<0.001, Kruskal-Wallis test with Dunn's Multiple Comparison *post hoc* test. (G) Summary data for oxytocin-induced EPSC amplitude changes in the presence of OxtR antagonist (L368,899), PLC blocker (U73122), and V1aR antagonist (SR49059). Wilcoxon signed rank test vs. baseline, ***p<0.001. (H) Left: Schematic of viral transduction strategy using AAV9-DIO-ChR2-mCherry to express ChR2 in PVN oxytocinergic neurons. Right, top: ChR2-mCherry (red) expression in PVN. Right, bottom: One example neuron identified as dopaminergic during whole-cell recording with Alexa Fluor 488 dye included in the internal solution (green). *Post hoc* immunofluorescent labeling against tyrosine hydroxylase (TH, grayscale) confirms its identity. (I) Left: Optical activation of oxytocin fibers in the VTA decreased evoked EPSC amplitude in DA neurons. Patterned light stimulation period is marked in blue. n = 9 neurons from six mice. Inset shows evoked EPSC traces from one VTA DA neuron before and after light stimulation, temporally matched to average data. Black dashed line shows EPSC amplitude of the example neuron at t = 120 s, 30 s after the end of optogenetic stimulation train. Right: Summary data for light stimulation-induced EPSC amplitude changes with and without OxtR antagonist (L368,899). n = 6 neurons from four mice in the presence of OxtR antagonist. **p<0.01, Wilcoxon signed rank test vs. baseline. ###p<0.001, Mann-Whitney test. Error bars reflect SEM. (J) Left: example traces of evoked EPSCs from one VTA DA neuron during baseline (blue) and after 30 s long light stimulation of PVN oxytocin fibers. Right: EPSC amplitudes recorded from one neuron (same as Left) during baseline and after light stimulation. (K) Summary data for light stimulation-induced EPSC amplitude changes 10–20 min and 20–30 min after light stimulation. *p<0.05, **p<0.01, Kruskal-Wallis test with Dunn's Multiple Comparison *post hoc* test, n = 6 neurons from four mice.
DOI: https://doi.org/10.7554/eLife.33892.003

The following source data and figure supplements are available for figure 1:

**Source data 1.** Summary tables of evoked EPSC amplitudes from individual VTA DA neurons in response to oxytocin-signaling related pharmacological agents and optogenetic stimulation of oxytocin fibers.
DOI: https://doi.org/10.7554/eLife.33892.006

**Figure supplement 1.** Characterization of targeted VTA subregions and physiological properties of recorded VTA DA neurons.
DOI: https://doi.org/10.7554/eLife.33892.004

**Figure supplement 2.** Optical stimulation of oxytocinergic fibers in the VTA modulates evoked EPSCs in DA neurons.
DOI: https://doi.org/10.7554/eLife.33892.005

*Xiao et al., 2017*). To determine whether oxytocin modulates evoked EPSCs by binding to V1aR we used a potent and selective V1aR antagonist (10 µM SR49059) in the bath solution. Under these conditions, oxytocin application decreased the amplitude of the evoked EPSCs to the same degree as in the baseline condition (80.94 ± 2.33% of baseline, p<0.001, Wilcoxon matched-pairs signed rank test, *n* = 9 neurons from six mice) (*Figure 1E and G*, *Figure 1—source data 1*).

To determine whether endogenous oxytocin release is sufficient to regulate synaptic transmission in VTA DA neurons, we relied on the *Oxt*[i-Cre] mice injected with a Cre-dependent rAAV expressing a fusion of ChR2 and the red fluorescent protein mCherry (ChR2-mCherry) into the PVN (*Figure 1H*) (*Xiao et al., 2017*). PVN oxytocin neuron projections that co-release glutamate have been reported in the parabrachial nucleus (*Ryan et al., 2017*) and in brainstem vagal neurons (*Piñol et al., 2014*). To evaluate potential co-release of glutamate in the VTA, we used single 10 ms-long light pulses to optically stimulate PVN oxytocin fibers while recording VTA DA neurons. All recorded neurons were validated using *post hoc* immunolabeling against tyrosine hydroxylase (TH), the rate-limiting enzyme for canonical dopamine synthesis (*Figure 1H*) (*Xiao et al., 2017*). In contrast to parabrachial and vagal neurons, no fast responses in VTA DA cells were observed (light-evoked response change −0.556 ± 0.285 pA, p=0.303, Wilcoxon matched-pairs signed rank test; *n* = 16 neurons from five mice). We then used trains of 10 ms-long light pulses at 20 Hz delivered in 30 second-long bursts to evoke axonal release of oxytocin directly in the acute slices from VTA (*Knobloch et al., 2012*; *Xiao et al., 2017*). Electrically evoked EPSCs were recorded before and after optical stimulation of oxytocin-positive axons (*Figure 1I*, *Figure 1—source data 1*, *Figure 1—figure supplement 2*).

Optical stimulation of oxytocin fibers depolarized VTA DA neurons in voltage-clamp recordings, evidenced by a small but consistent change in holding current (2.955 ± 0.871 pA, p<0.01, Wilcoxon signed rank test, $n$ = 15 neurons from eight mice), consistent with prior observations of tonic activity changes in VTA DA neurons in response to oxytocin. As for bath application experiments, we observed that the amplitude of evoked EPSC in VTA DA neurons was decreased, evident 30 s after optical stimulation (61.71 ± 4.39% of baseline, p<0.01, Wilcoxon matched-pairs signed rank test, $n$ = 9 neurons from six mice) (**Figure 1I**, **Figure 1—figure supplement 2**). This repeatable effect was abolished in the presence of OxtR antagonist (L368,899) (114.30 ± 8.04% of baseline, p=0.156, Wilcoxon matched-pairs signed rank test, $n$ = 6 neurons from four mice) (**Figure 1I**, **Figure 1—figure supplement 2**). We also observed that 30 s long light activation of oxytocinergic fibers in VTA is sufficient to lead to a relatively long-lasting decrease in the amplitude of EPSCs (5–15 min after light stimulation: 69.22 ± 6.72% of baseline; 15–25 min after light stimulation: 57.77 ± 7.01% of baseline, p<0.01, Kruskal-Wallis test with Dunn's Multiple Comparison *post hoc* test, $n$ = 6 neurons from four mice) (**Figure 1J–K**, **Figure 1—source data 1**).

## Oxytocin dampens excitatory synaptic transmission through presynaptic mechanisms

Both presynaptic neurotransmitter release and postsynaptic receptor properties contribute to the strength of synaptic transmission. We first assessed whether oxytocin regulates excitatory synaptic transmission by decreasing glutamate release. To accomplish this we started by measuring the paired-pulse ratio (PPR) following two electrical stimuli (50 ms interstimulus interval, ISI). Oxytocin application significantly increased PPR at a 50 ms ISI but not 80 or 100 ms ISI (Baseline, 1.120 ± 0.057; Oxt, 1.721 ± 0.116; p<0.01, Wilcoxon matched-pairs signed rank test, $n$ = 8 neurons from six mice) (**Figure 2A–B**, **Figure 2—source data 1**, **Figure 2—figure supplement 1**), indicating that oxytocin decreases the probability of glutamate release. Blocking either OxtR or the PLC pathway abolished the effect on PPR. However, in the presence of V1aR antagonist, oxytocin application still significantly increased PPR (Baseline, 0.948 ± 0.076; Oxt, 1.326 ± 0.149; p<0.01, Wilcoxon matched-pairs signed rank test, $n$ = 8 neurons from five mice) (**Figure 2A–B**). Optical activation of oxytocin-positive axons in VTA was sufficient to increase PPR 30 s after light stimulation, consistent with the effect of light stimulation on EPSC amplitude changes (**Figure 2C**, **Figure 2—figure supplement 2**). These light-evoked changes in PPR, as for the amplitude attenuation, were likewise blocked by the OxtR antagonist (Control: Baseline, 1.220 ± 0.127; Light, 1.634 ± 0.195; p<0.01, Wilcoxon matched-pairs signed rank test, $n$ = 9 neurons from six mice; L368,899: Baseline, 1.454 ± 0.232; Light, 1.147 ± 0.113; p=0.156, Wilcoxon matched-pairs signed rank test, $n$ = 6 neurons from four mice) (**Figure 2D**, **Figure 2—source data 1**, **Figure 2—figure supplement 2**). To further probe potential effects of oxytocin on presynaptic glutamate release, we evaluated spontaneous EPSCs (sEPSC) of VTA DA neurons. Bath application of oxytocin decreased sEPSC frequency (Baseline, 2.328 ± 0.308 Hz; Oxt, 1.679 ± 0.237 Hz; p<0.05, Wilcoxon matched-pairs signed rank test, $n$ = 6 neurons from five mice), but had no significant effect on sEPSC amplitude (Baseline, 7.741 ± 1.060 pA; Oxt, 8.830 ± 1.389 pA; p=0.094, Wilcoxon matched-pairs signed rank test) (**Figure 2E–F**, **Figure 2—source data 1**). Together, these results are consistent with the interpretation that oxytocin reduces the probability of glutamate release in excitatory inputs to VTA DA neurons.

While oxytocin appears to alter presynaptic glutamate release, the experiments above do not exclude the possibility of oxytocin also regulating postsynaptic receptor properties. In order to explicitly test this possibility, we carried out two-photon glutamate uncaging experiments to directly stimulate individual spiny protrusions on VTA DA dendrites using bath-applied MNI-L-Glutamate (**Figure 2G**). VTA DA neurons from *Slc6a3*[i-Cre]; Ai14 mice were filled with Alexa Fluor 488 through the recording pipette. Dendritic spines were visualized using laser-scanning 2-photon imaging at 910 nm, and glutamate was uncaged via two brief pulses (0.5 ms duration, 50 ms ISI) of 720 nm light with a second mode-locked Ti-Sapphire laser. Uncaging-evoked EPSCs (uEPSCs) were recorded at the soma at a holding potential of −70 mV in the presence of 10 µM GABA(A) receptor antagonist. The amplitude of uEPSCs evoked by the first light pulse recorded in the baseline condition was similar to that recorded in the presence of 1 µM oxytocin (**Figure 2H–I**, **Figure 2—source data 1**). Moreover, oxytocin did not alter the PPR of uEPSCs, evaluated using both paired measurements on single spines following oxytocin flow-in (five dendritic spines from four neurons in three mice) or in

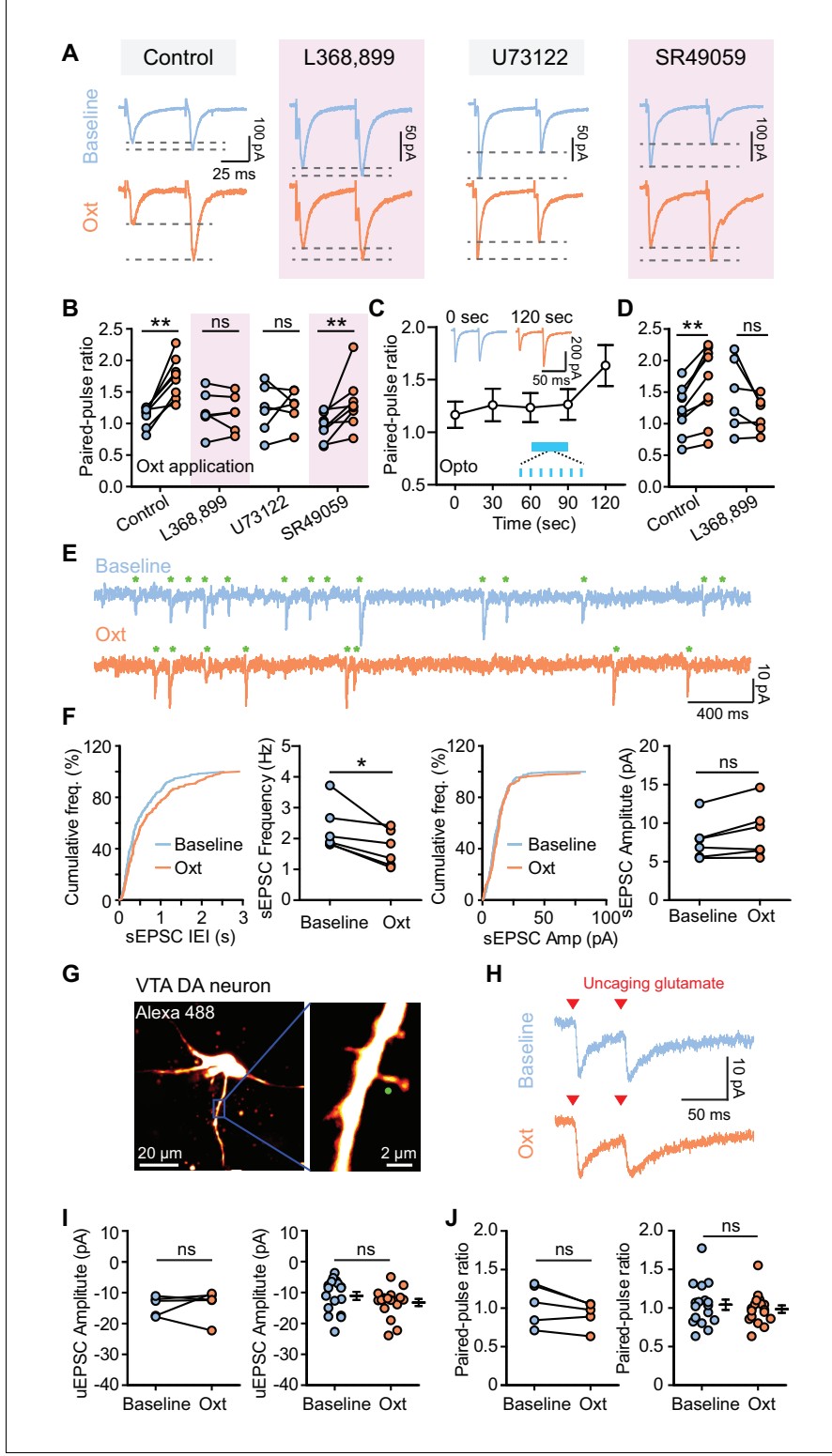

**Figure 2.** Oxytocin dampens excitatory synaptic transmission by presynaptic mechanisms. (**A**) Traces from VTA DA neurons showing pairs of EPSCs (50 ms ISI) during baseline (top) and following oxytocin application (bottom). From left to right, example traces reflect: the control condition with no additional compounds, the OxtR antagonist L368,899, the PLC blocker U73122, and the V1aR antagonist SR49059. Traces for each neuron are normalized to the amplitude of the first evoked response in the baseline condition. (**B**) Summary results show paired-pulse ratio during baseline and following oxytocin application in control condition, in the presence of OxtR

*Figure 2 continued on next page*

*Figure 2 continued*

antagonist L368,899, PLC blocker U73122, and V1aR antagonist SR49059 separately. **p<0.01, Wilcoxon matched-pairs signed rank test. For each condition respectively, n = 8 neurons from six mice, 6 neurons from five mice, 6 neurons from four mice, and 8 neurons from five mice. (C) Summary results show PPR change of the evoked EPSC of VTA DA neurons in response to light stimulation (marked in blue). n = 9 neurons from six mice. Insets show example pairs of EPSCs (50 ms ISI) from one VTA DA neuron at 0 s (before light stimulation) and 120 s (after light stimulation). (D) Summary data for light stimulation-induced EPSC PPR changes with and without OxtR antagonist (L368,899). **p<0.01, Wilcoxon matched-pairs signed rank test, n = 9 neurons from six mice for control condition, n = 6 neurons from four mice in the presence of OxtR antagonist. (E) Example traces show spontaneous EPSCs (sEPSCs, marked with green stars) during baseline (top) and following oxytocin application (bottom). (F) Left: distributions of sEPSC inter-event intervals (IEI) during baseline and following oxytocin application, with summary data for sEPSC frequency. Right: distributions of sEPSC amplitudes and summary data for sEPSC amplitude during baseline and oxytocin application. *p<0.05, Wilcoxon matched-pairs signed rank test, n = 6 neurons from five mice. (G) Left: projection of a z-stack from a VTA DA neuron filled with Alexa Fluor 488 acquired on a two-photon laser-scanning microscope. Right: higher magnification image of the dendrite. Green circle marks the uncaging spot. (H) Example traces of paired uncaging-evoked EPSCs (uEPSCs) at 50 ms ISI, from one dendritic spine during baseline condition (top) and oxytocin application (bottom). Red triangles mark uncaging time-points. (I) Left: summary data for uEPSC amplitudes in a group of dendritic spines from VTA DA neurons during baseline and following oxytocin application. p=0.813, Wilcoxon matched-pairs signed rank test, n = 5 dendritic spines from 4 neurons in three mice. Right: summary data for all uEPSC amplitudes across conditions. p=0.191, Mann-Whitney test, n = 17 dendritic spines for each condition from 16 neurons in four mice. (J) Same as (I), but for paired-pulse ratio. Error bars reflect SEM.

DOI: https://doi.org/10.7554/eLife.33892.007

The following source data and figure supplements are available for figure 2:

**Source data 1.** Summary tables of paired pulse ratios of evoked EPSCs, inter-event intervals of spontaneous EPSCs, and amplitudes of uncaging-evoked EPSCs.

DOI: https://doi.org/10.7554/eLife.33892.010

**Figure supplement 1.** Oxytocinergic modulation of paired-pulse ratio in VTA DA neurons.

DOI: https://doi.org/10.7554/eLife.33892.008

**Figure supplement 2.** Optical stimulation of oxytocinergic fibers in VTA modulates paired-pulse ratio of evoked EPSCs in DA neurons.

DOI: https://doi.org/10.7554/eLife.33892.009

group data (17 dendritic spines for each condition from 16 neurons in four mice) (*Figure 2H and J*, *Figure 2—source data 1*). These results suggest that oxytocin does not modulate glutamatergic receptor properties of VTA DA neurons, confirming its selective presynaptic effects on synaptic transmission.

## Retrograde endocannabinoid signaling underlies oxytocinergic modulation of excitatory synaptic transmission

Previous studies demonstrate that increasing the activity of VTA DA neurons triggers these neurons to release endocannabinoids, which can activate presynaptic cannabinoid CB1 receptors to regulate neurotransmitter release (*Melis et al., 2004*; *Oleson and Cheer, 2012*; *Riegel and Lupica, 2004*). Since oxytocin weakens evoked EPSCs in VTA DA neurons by reducing presynaptic glutamate release, retrograde endocannabinoid signaling could provide a possible mechanism for the observed effects. We also observed that the activation of CB receptors with a selective agonist (WIN55212-2) decreases evoked EPSC amplitude (79.77 ± 1.75% of baseline, n = 7 neurons from five mice, p<0.01, Wilcoxon signed rank test) and increases PPR (Baseline, 0.780 ± 0.047; WIN, 0.947 ± 0.061; n = 7 neurons from five mice, p<0.01, Wilcoxon matched-pairs signed rank test) (*Figure 3A,G*, *Figure 3—source data 1*, *Figure 3—figure supplement 1*). In the presence of WIN55212-2, occluded by CB1 receptor activation, oxytocin application failed to modulate the amplitude and PPR of evoked EPSCs in VTA DA neurons (*Figure 3B,G*, *Figure 3—source data 1*, and *Figure 3—figure supplement 1*). Conversely, blocking cannabinoid CB1 receptors with a selective, high affinity CB1 receptor inverse agonist (AM251) or antagonist (CP945598) abolished oxytocinergic modulation of EPSC amplitude (AM251: 97.12 ± 2.97% of baseline, p=0.438, Wilcoxon signed rank test, n = 6 neurons from three mice; CP945598: 98.76 ± 2.70% of baseline, p=0.688, Wilcoxon signed rank test,

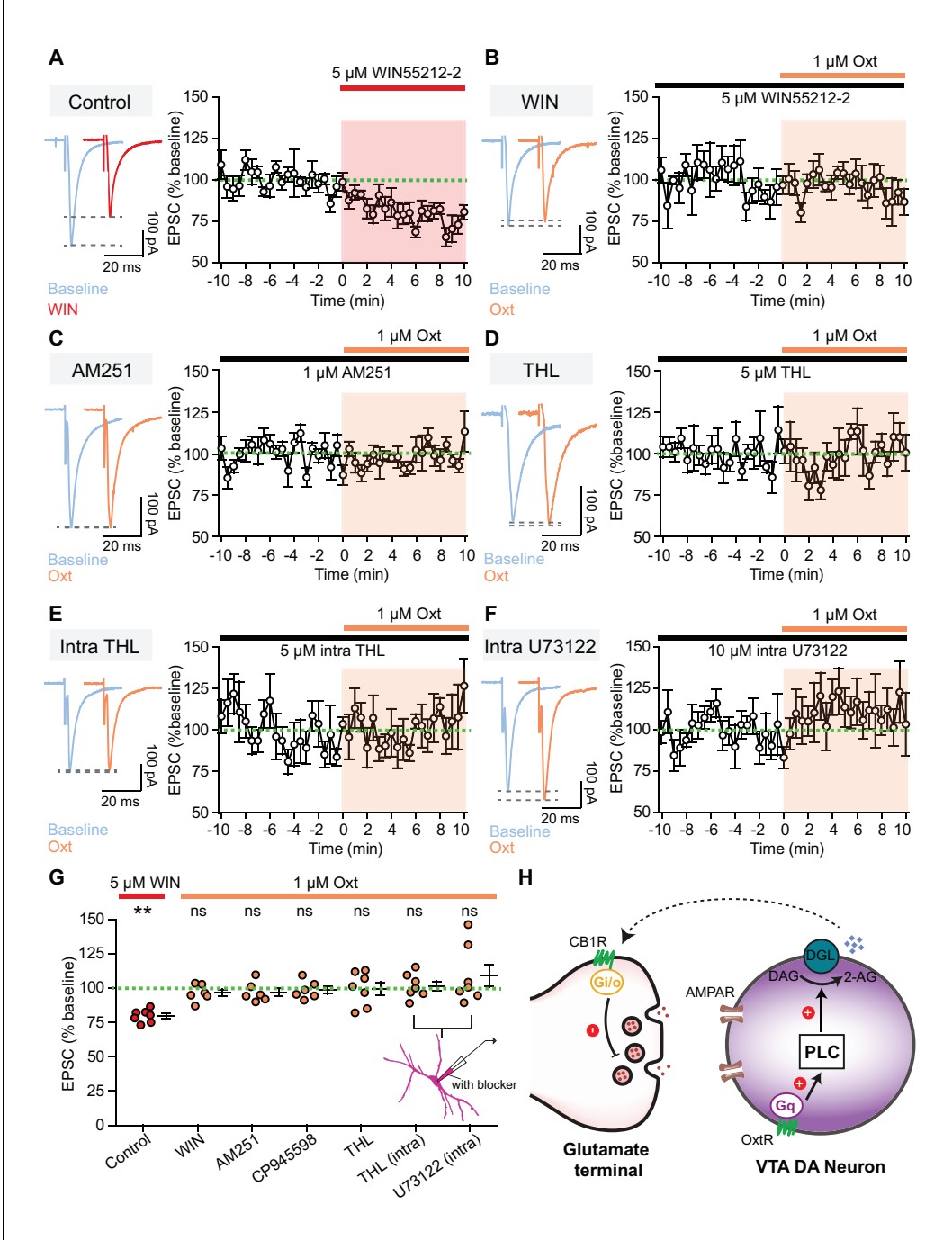

**Figure 3.** Retrograde endocannabinoid signaling underlies oxytocinergic modulation of excitatory synaptic transmission. (**A**) Left: example traces of evoked EPSCs from one VTA DA neuron during baseline and following 5 µM CB1R agonist WIN55212-2 application. Right: summary data for EPSC amplitude over time, normalized to baseline. (**B**) Left: example traces of evoked EPSCs from one VTA DA neuron during baseline and following 1 µM oxytocin application in the presence of CB1R agonist WIN55212-2. Right: summary data for EPSC amplitude over time, normalized to baseline. (**C**) Same as (**B**), but in the presence of CB1R inverse agonist AM251. (**D**) Same as (**B**), but in the presence of orlistat (THL), a blocker of 2-AG synthesizing enzyme diacylglycerol lipase α (DGLα). (**E**) Same as (**B**), but with THL in the internal solution to block 2-AG synthesis. (**F**) Same as (**B**), but with U73122 in the internal solution to inhibit PLC activity. (**G**) Summary data for EPSC amplitude changes induced by CB1R agonist WIN55212-2 and oxytocin in the presence of WIN55212-2, CB1R inverse agonist AM251, CB1R antagonist CP945598, orlistat (THL), intracellular THL, and intracellular U73122, separately. **p<0.01, Wilcoxon signed rank

*Figure 3 continued on next page*

*Figure 3 continued*
test vs. baseline. For each condition respectively, n = 7 neurons from five mice, six neurons from four mice, six neurons from three mice, six neurons from four mice, six neurons from four mice, seven neurons from four mice, and seven neurons from four mice. Error bars reflect SEM. (H) Schematic summary of the mechanism underlying direct oxytocinergic regulation of excitatory synaptic transmission in VTA DA neurons. Oxytocin binds to $G_q$-coupled OxtR, activating the phospholipase C (PLC) pathway. PLC cleaves phosphatidylinositol 1,4,5-bisphosphate into diacylglycerol (DAG) and inositol 1,4,5-trisphosphate, and DAG is subsequently converted into 2-AG by DAG lipase (DGL). 2-AG released by VTA DA neurons binds to presynaptic $G_{i/o}$-coupled CB1R, decreasing glutamate release.

DOI: https://doi.org/10.7554/eLife.33892.011

The following source data and figure supplement are available for figure 3:

**Source data 1.** Summary tables of evoked EPSC amplitudes in response to endocannabinoid-signaling related pharmacological agents.

DOI: https://doi.org/10.7554/eLife.33892.013

**Figure supplement 1.** Endocannabinoid signaling underlies oxytocinergic regulation of the amplitude and PPR of evoked EPSCs in VTA DA neurons.

DOI: https://doi.org/10.7554/eLife.33892.012

n = 6 neurons from four mice) and of the PPR of evoked EPSCs (*Figure 3C*, *Figure 3G*, *Figure 3—source data 1*, and *Figure 3—figure supplement 1*).

The two major mammalian endocannabinoids are the bioactive lipid 2-arachidonoylglycerol (2-AG) and anandamide, and previous evidence suggests that 2-AG is a primary endocannabinoid synthesized and released by VTA DA neurons (*Gantz and Bean, 2017*; *Mátyás et al., 2008*; *Merrill et al., 2015*). PLC cleaves phosphatidylinositol 1,4,5-bisphosphate into diacylglycerol (DAG) and inositol 1,4,5-trisphosphate; DAG is subsequently converted into 2-AG by DAG lipase (*Di Marzo et al., 1998*; *Piomelli, 2003*). To determine the identify and function of endocannabinoids in oxytocinergic modulation of synaptic transmission in VTA DA neurons, we blocked the 2-AG synthesizing enzyme DAG lipase α (DGLα) with orlistat (THL) (*Gantz and Bean, 2017*; *Tanimura et al., 2010*). Brain slices were treated with 1 µM THL for ~1 hr before and during whole-cell recording. Blocking 2-AG synthesis with THL abolished oxytocinergic effects on the amplitude of evoked EPSCs (99.40 ± 5.54% of baseline, p=0.938, Wilcoxon signed rank test, n = 6 neurons from four mice) (*Figure 3D,G*, *Figure 3—source data 1*) and on the PPR (*Figure 3—figure supplement 1*). Since bath application of orlistat cannot distinguish cell-autonomous from circuit-level effects of inhibiting 2-AG synthesis, we next carried out experiments using intracellular inhibitors of PLC or 2-AG synthesis (U73122 or THL) in internal recording solution. Intracellular blockade of either the PLC pathway or 2-AG synthesis was sufficient to abolish oxytocinergic effects on the amplitude of evoked EPSCs and PPR (*Figure 3E–G*, *Figure 3—source data 1*, *Figure 3—figure supplement 1*). Therefore, in the VTA, oxytocin activates the PLC/DAG lipase pathway to increase 2-AG release, weakening glutamatergic synaptic transmission in a cell-autonomous manner (*Figure 3H*).

## Oxytocin modulates the magnitude of short- and long-term synaptic plasticity of excitatory synaptic transmission

Synaptic plasticity of glutamatergic transmission in VTA DA neurons provides a critical substrate relevant to learning and addiction (*Chen et al., 2010*; *Kauer and Malenka, 2007*; *Niehaus et al., 2009*), and endocannabinoids play a pivotal role in synaptic plasticity by depressing synaptic transmission on short- and long-term scales (*Gerdeman et al., 2002*; *Di Marzo et al., 1998*). Given the observed effects of oxytocin on endocannabinoid-dependent regulation of excitatory synaptic transmission and PPR, we further probed the potential for oxytocinergic modulation of plasticity-inducing stimuli in VTA DA neurons. Ten electrical stimuli at 50 ms ISI were used to induce short-term synaptic plasticity. The amplitude of EPSCs in most recorded VTA DA neurons (6/8 neurons) decreased to a steady state when stimulated using this protocol in the control condition. During oxytocin perfusion, the magnitude of EPSC attenuation decreased, as measured by the ratio of last to first evoked EPSC amplitude. This decrement is accounted for by the drop in the amplitude of EPSCs evoked by the first two electrical stimulations (*Figure 4A–B*, *Figure 4—source data 1*). Oxytocin perfusion significantly increased the ratio between the amplitudes of the last and first EPSCs in a stimuli train

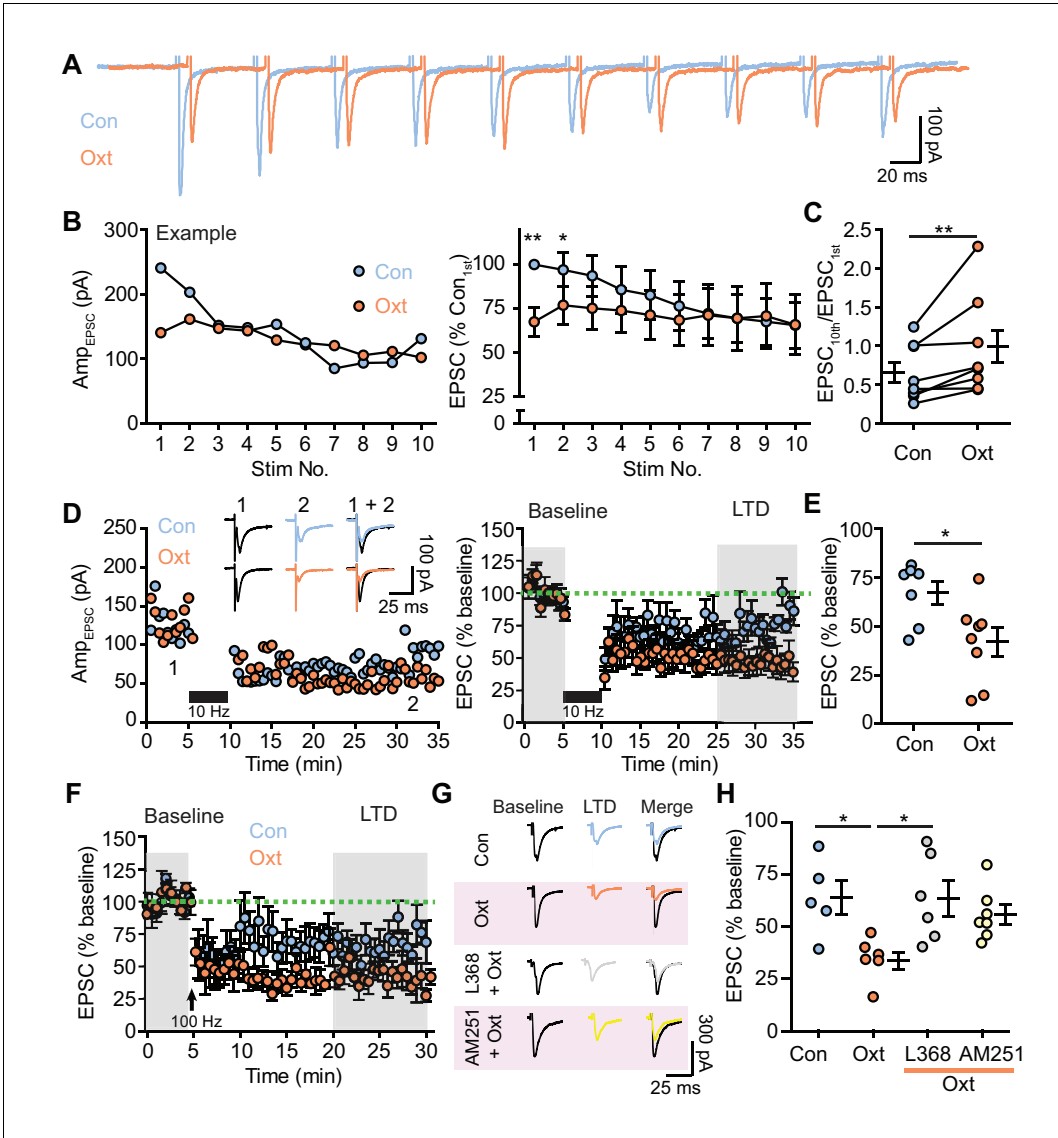

**Figure 4.** Oxytocin contributes to short and long-term synaptic plasticity of excitatory synaptic transmission. (**A**) Example voltage-clamp traces from one VTA DA neuron evoked by repetitive electrical stimulation (20 Hz, 10 pulses) in control condition (blue) and during oxytocin application (orange). (**B**) Left: evoked EPSC amplitudes from the example neuron shown in (**A**) before and during oxytocin application. Right: summary results of normalized EPSC in control condition and following oxytocin application. EPSC amplitudes were normalized to the amplitude of the first EPSC in the control group. **$p<0.001$, *$p<0.05$, Wilcoxon matched-pairs signed rank test, n = 8 neurons from five mice. (**C**) Oxytocin increases the ratio between the 10th EPSC and the first EPSC within a train of stimuli. **$p<0.01$, Wilcoxon matched-pairs signed rank test, n = 8 neurons from five mice. (**D**) Oxytocin application adds to long-term depression (LTD) of EPSCs induced by 5 min, 10 Hz repetitive electrical stimulation (black bars). Left: EPSC amplitudes recorded from two representative neurons in the control condition and in the presence of oxytocin. Inset shows example traces from two neurons before and after LTD induction. Right: summary results show the normalized evoked EPSC amplitude across time in the control condition and in the presence of oxytocin. EPSC amplitudes were normalized based on a 5 min baseline recording (first shaded interval). n = 7 neurons from five mice for controls; eight neurons from six mice in the presence of oxytocin. (**E**) Summary data showing normalized EPSC amplitude after LTD induction (second shaded interval indicated in (**D**)). *$p<0.05$, Mann-Whitney test, n = 7 and 8 neurons for control condition and in the presence of oxytocin. (**F**) Same as (**D**), but with LTD induction using 1 s long periods of 100 Hz stimulation, repeated four times with 10 second-long inter-train intervals (indicated by arrows). n = 5 neurons from five mice for controls, and six neurons from six mice for oxytocin condition. (**G**) Example traces before and after LTD induction in control condition, in the presence of oxytocin, in the presence of L368,899 and oxytocin, and in the presence of AM251 and oxytocin. (**H**) Summary data

*Figure 4 continued on next page*

*Figure 4 continued*

showing normalized EPSC amplitude after LTD induction (second shaded interval indicated in (**F**)) in the control condition, in the presence of oxytocin, in the presence of oxytocin and OxtR antagonist L368, 899, and in the presence of oxytocin and CB1 receptor inverse agonist AM251. *p<0.05, Kruskal-Wallis test with Dunn's Multiple Comparison *post hoc* test, n = 5 neurons for control, n = 6 neurons for oxytocin condition and oxytocin and L368, 899 condition, n = 7 neurons for oxytocin and AM251 condition.

DOI: https://doi.org/10.7554/eLife.33892.014

The following source data and figure supplements are available for figure 4:

**Source data 1.** Summary tables of amplitudes of repetitively evoked EPSCs in response to oxytocin and endocannabinoid-signaling related pharmacological agents.

DOI: https://doi.org/10.7554/eLife.33892.017

**Figure supplement 1.** Oxytocin regulates presynaptic neurotransmitter release during LTD.

DOI: https://doi.org/10.7554/eLife.33892.015

**Figure supplement 2.** Oxytocin modulates EPSC amplitude during LTD through oxytocin and endocannabinoid receptors.

DOI: https://doi.org/10.7554/eLife.33892.016

---

(Control: 0.655 ± 0.131; Oxt: 0.974 ± 0.228; p<0.05, Wilcoxon matched-pairs signed rank test, *n* = 8 neurons from five mice) (*Figure 4C*, *Figure 4—source data 1*).

Given the relatively long-lasting, presynaptic effects of oxytocin on glutamatergic synaptic transmission, we questioned whether this modulatory mechanism can contribute to long-term depression (LTD) of excitatory inputs to VTA DA neurons, an important plasticity form for these cells. We used 10 Hz repetitive electrical stimulation (5 min) or 100 Hz stimulation (1 second-long period, repeated 4 times with 10 second-long inter-train intervals) in separate experiments to induce LTD (*Good and Lupica, 2010*; *Kreitzer and Malenka, 2005*; *Pan et al., 2008*). In the control condition, EPSC amplitude decreased to 64.22 ± 4.70% of baseline following 10 Hz stimulation (seven neurons from five mice) (*Figure 4D–E*, *Figure 4—source data 1*) and to 64.00 ± 8.19% of baseline following 100 Hz stimulation (five neurons from five mice) (*Figure 4F–G*, *Figure 4—source data 1*). In the presence of oxytocin, LTD magnitude induced by either stimulation protocol was substantially increased (10 Hz: 39.42 ± 6.19% of baseline, eight neurons from six mice, p<0.05, Mann-Whitney test vs. control; 100 Hz: 34.77 ± 4.18% of baseline, six neurons from six mice, p<0.05, Mann-Whitney test vs. control) (*Figure 4D–H*, *Figure 4—source data 1*). Input resistance of VTA DA neurons was similar before and after LTD induction and was not altered by oxytocin (*Figure 4—figure supplement 1A and C*). The PPR of EPSCs remained unchanged through LTD induction in the control condition (*Figure 4—figure supplement 1B and D*), but increased following LTD induction in the presence of oxytocin for both stimulation protocols (*Figure 4—figure supplement 1B and D*).

To confirm the involvement of OxtRs and CB1Rs in the additive presynaptic component to LTD in the presence of oxytocin, brain slices were treated with 10 µM L368,899 or 5 µM AM251 for ~30 min before and during whole-cell recording. In the presence of oxytocin and either OxtR antagonist L368,899 or the CB1 receptor inverse agonist AM251, LTD magnitude induced by 100 Hz stimulation was similar to that in the control condition and smaller than the LTD magnitude in the presence of oxytocin (L368,899: 63.46 ± 8.48% of baseline, six neurons from four mice; AM251: 55.77 ± 4.69% of baseline, seven neurons from four mice, p<0.05, Kruskal-Wallis test with Dunn's Multiple Comparison *post hoc* test) (*Figure 4F–H*, *Figure 4—source data 1*, and *Figure 4—figure supplement 2*). Together, oxytocin modulates both short- and long-term synaptic plasticity in VTA DA neurons by presynaptic mechanisms mediated by endocannabinoids.

## Oxytocin modulation of excitatory synaptic transmission is pathway-specific

While brain regions that send glutamatergic projections to the VTA – but not the VTA DA neurons themselves – are known to express the CB1 receptors (*Mátyás et al., 2008*), it remains unknown whether specific groups of VTA-projecting neurons selectively express this receptor. To address this we carried out combined retrograde labeling using green retrobeads (GRB) injected into the VTA and quantitative multi-channel fluorescence in situ hybridization (FISH) assays for mRNA of CB1 receptor gene (*Cnr1*) in several brain regions. Although there are not many published reports using

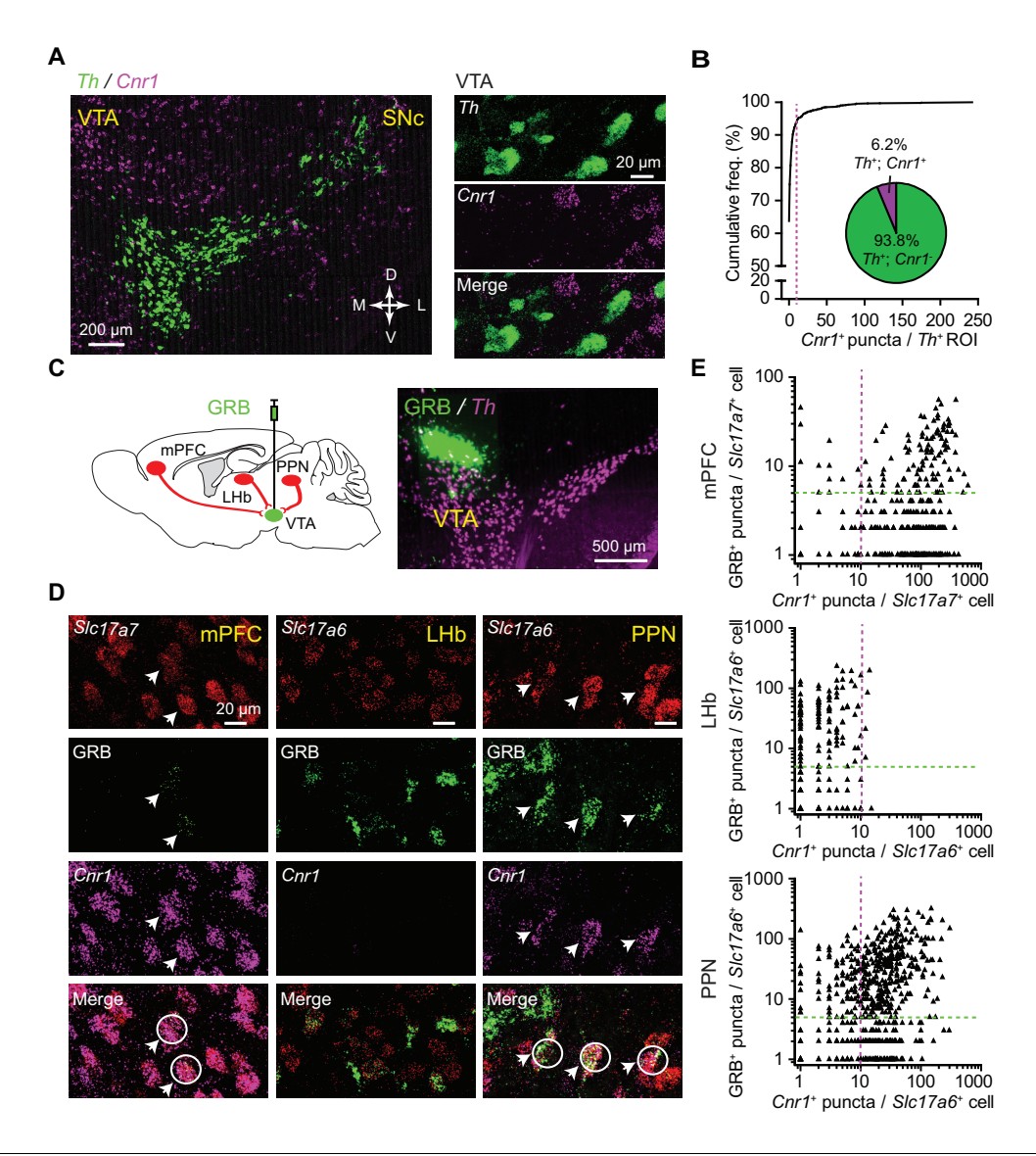

**Figure 5.** CB1 receptor transcripts selectively express in a subset of VTA DA neuron-targeting glutamatergic afferents. (**A**) Left: fluorescence in situ hybridization (FISH) images of *Th* and CB1R (*Cnr1*) in midbrain VTA region. Green, *Th*; magenta, *Cnr1*. Right: confocal image of *Th* and *Cnr1* in the VTA, showing separate channels and overlay. (**B**) Quantitative analysis of *Th* and *Cnr1* co-expression by FISH in the VTA. Magenta line marks the cutoff used to classify a $Th^+$ cell as $Cnr1^+$ (>10 $Cnr1^+$ puncta). Insert shows the proportion of $Th^+$ neurons classified as $Cnr1^+$ and $Cnr1^-$. n = 1258 $Th^+$ VTA cells from three mice. (**C**) Left: schematic showing combined retrograde labeling and FISH analysis. Right: example image of a green retrobead injection (GRB) in the VTA with follow up FISH for *Th*. (**D**) Left: confocal image of *Slc17a7*, GRB and *Cnr1* in the medial prefrontal cortex (mPFC), showing separate channels and overlay. Arrows and white circles mark glutamatergic neurons co-localizing GRB signal and *Cnr1*. Middle: confocal image of *Slc17a6*, GRB and *Cnr1* in the lateral habenular nucleus (LHb). Right: same as middle, but in the pedunculopontine nucleus (PPN). (**E**) Top: Quantitative analysis of *Slc17a7*, *Cnr1*, and GRB co-expression by FISH in the mPFC. Magenta and green lines mark the cutoffs used to classify a cell as $Cnr1^+$ or $GRB^+$. 143 $GRB^+/Slc17a7^+/Cnr1^+$ neurons of 188 $GRB^+/Slc17a7^+$ neurons from three mice. Middle: Same as (Top), but in LHb and labeled glutamatergic neurons with *Slc17a6*. 4 $GRB^+/Slc17a6^+/Cnr1^+$ neurons of 178 $GRB^+/Slc17a6^+$ neurons from two mice. Bottom: Same as (Top), but in PPN and labeled glutamatergic neurons with *Slc17a6*. 306 $GRB^+/Slc17a6^+/Cnr1^+$ neurons of 401 $GRB^+/Slc17a6^+$ neurons from three mice.

DOI: https://doi.org/10.7554/eLife.33892.018

The following source data and figure supplement are available for figure 5:

*Figure 5 continued on next page*

*Figure 5 continued*

**Source data 1.** Summary tables of the number of FISH and green retrobead fluorescent puncta in each analyzed cell in regions of interest.
DOI: https://doi.org/10.7554/eLife.33892.020
**Figure supplement 1.** CB1 receptor transcripts expression in arcuate nucleus $Th^+$ neurons and glutamatergic neurons projecting to VTA.
DOI: https://doi.org/10.7554/eLife.33892.019

the combination of retrobead injections with FISH assays, we found this combination robust and useful for evaluating transcripts within pathway-specific populations of neurons. Confirming prior reports (*Mátyás et al., 2008*), we did not find substantial CB1 receptor expression in VTA DA neurons (*Figure 5A–B*, *Figure 5—source data 1*), even though DA neurons in the arcuate nucleus do express the CB1 receptor (*Figure 5—figure supplement 1A*). DA neurons were identified by labeling for tyrosine hydroxylase (*Th*), and few $Th^+$ neurons in the VTA co-localized with $Cnr1^+$ puncta (<7%, n = 1258 $Th^+$ neurons from three mice) (*Figure 5B*). Next, we quantified co-localization of GRB and *Cnr1* signals with glutamatergic neurons, identified by the expression of *Slc17a7* (*Vglut1*) or *Slc17a6 (Vglut2)* (*Figure 5C*). We focused on three well-studied regions that project to the VTA: medial prefrontal cortex (mPFC), lateral habenular nucleus (LHb), and pedunculopontine nucleus (PPN) (*Beier et al., 2015*; *Good and Lupica, 2010*). In the mPFC, *Slc17a7* probe was used to label glutamatergic neurons; there, the vast majority of glutamatergic neurons projecting to VTA (GRB$^+$/$Slc17a7^+$) co-localized with strong *Cnr1* signal (143 GRB$^+$/$Slc17a7^+$/$Cnr1^+$neurons of 188 GRB$^+$/$Slc17a7^+$ neurons from three mice) (*Figure 5D–E*, *Figure 5—source data 1*, *Figure 5—figure supplement 1B*). We labeled glutamatergic neurons in LHb and PPN using a *Slc17a6* probe, observing extensive overlap between GRB signal and *Slc17a6* puncta in both of these regions. However, LHb had almost no *Cnr1* expression, so triple-labeled neurons were rare (4 GRB$^+$/$Slc17a6^+$/$Cnr1^+$neurons of 178 GRB$^+$/$Slc17a6^+$ neurons from two mice) (*Figure 5D–E*, *Figure 5—source data 1*, *Figure 5—figure supplement 1B*). In contrast, over 70% of GRB$^+$/$Slc17a6^+$ neurons in the PPN co-localized with *Cnr1* signal (306 GRB$^+$/$Slc17a6^+$/$Cnr1^+$neurons of 401 GRB$^+$/$Slc17a6^+$ neurons from three mice) (*Figure 5D–E*, *Figure 5—source data 1*, *Figure 5—figure supplement 1B*).

The observation that CB1 receptor transcripts express in a subset of VTA glutamatergic afferents leads to a strong prediction that oxytocin modulates only those inputs, allowing CB1R-negative inputs to pass unchanged. To test this, we turned to optogenetic stimulation of specific glutamatergic afferents, using $Slc17a6^{i-Cre}$ mice, with a Cre-dependent rAAV expressing ChR2-mCherry delivered into the mPFC, LHb, or PPN, respectively (*Figure 6A*). After 4–5 weeks, 2 ms 470 nm light was used to activate glutamatergic axons in the VTA while we recorded light-evoked EPSCs in VTA DA neurons (validated with *post hoc* TH immunolabeling, *Figure 6A*). For neurons that responded to light stimulation with time-locked EPSCs, 1 µM oxytocin was applied following >10 min long baseline acquisition. Following mPFC targeting, 9/30 VTA DA neurons from four mice responded to light stimulation, and oxytocin application attenuated the amplitude of light-evoked EPSCs (73.35 ± 3.83% of baseline, p<0.05, Wilcoxon signed rank test, *n* = 6 neurons from four mice) (*Figure 6B–C*, *Figure 6—source data 1*, *Figure 6—figure supplement 1*). After PPN transduction, 8/10 VTA neurons from two mice responded to light stimulation, and oxytocin also significantly dampened light-evoked EPSCs (82.55 ± 0.76% of baseline, p<0.05, Wilcoxon signed rank test, *n* = 7 neurons from two mice) (*Figure 6B–C*, *Figure 6—source data 1*, *Figure 6—figure supplement 1*). In contrast, with ChR2 virus targeted to the LHb, while 8/21 VTA DA neurons responded to light stimulation, oxytocin had no effect on the amplitude of evoked EPSCs (94.84 ± 2.27% of baseline, p=0.219, Wilcoxon signed rank test, *n* = 7 neurons from four mice) (*Figure 6B–C*, *Figure 6—source data 1*, *Figure 6—figure supplement 1*). These data likely indicate that distinct sources of glutamate converging on individual VTA DA neurons are differentially regulated by oxytocin based on CB1 receptor expression, but an alternative interpretation of these results is that VTA DA neurons receiving inputs from lateral habenula selectively lack OxtRs.

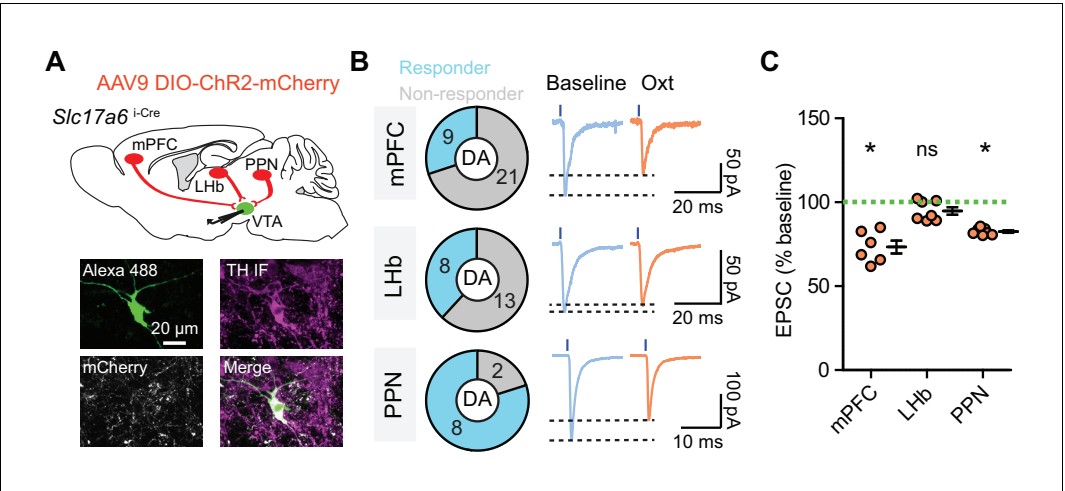

**Figure 6.** Oxytocin modulates pathway-specific excitatory synaptic transmission in VTA DA neurons. (**A**) Top: Schematic of viral transduction strategy using AAV9-DIO-ChR2-mCherry to express ChR2 in glutamatergic neurons in mPFC, LHb, and PPN, respectively. Bottom: One example neuron labeled with Alexa Fluor 488 dye (green) during recording, identified as dopaminergic with TH *post hoc* immunolabeling (magenta). mCherry[+] glutamatergic axons (gray) from the PPN are visible in the proximity of the dye-filled neuron. (**B**) Left: Quantification of VTA DA neurons receiving glutamatergic inputs from each brain region. Right: Examples of light-evoked EPSC traces of VTA DA neurons under baseline conditions and during oxytocin application with ChR2-mCherry virus delivered into mPFC, LHb, and PPN, respectively. Black dashed lines indicate the peak value of each trace. (**C**) Summary data for light-evoked EPSC amplitude at baseline and following oxytocin application with ChR2-mCherry virus delivered into mPFC, LHb, and PPN, respectively. *p<0.05, Wilcoxon signed rank test vs. baseline, n = 6 neurons from four mice for mPFC, seven neurons from four mice for LHb, and 7 neurons from two mice for PPN. Error bars reflect SEM.

DOI: https://doi.org/10.7554/eLife.33892.021

The following source data and figure supplement are available for figure 6:

**Source data 1.** Summary table of the effect of oxytocin on excitatory synaptic inputs from various regions to the VTA.
DOI: https://doi.org/10.7554/eLife.33892.023

**Figure supplement 1.** Oxytocin attenuates pathway-specific glutamatergic synaptic transmission in VTA DA neurons.
DOI: https://doi.org/10.7554/eLife.33892.022

## Discussion

Oxytocin has been previously reported to modulate the excitability of VTA DA neurons (*Hung et al., 2017*; *Tang et al., 2014*; *Xiao et al., 2017*), but how this peptide regulates synaptic transmission to these same DA neurons has not been investigated at the mechanistic level, precluding an integrative conceptual framework of peptidergic actions in midbrain DA systems. Here, we find that oxytocin inhibits excitatory synaptic transmission via OxtR and retrograde endocannabinoid signaling, modulating both short- and long-term plasticity in a subset of VTA DA neurons through presynaptic mechanisms. Furthermore, we observe a heterogeneity of endocannabinoid CB1 receptor expression among the glutamatergic regions projecting to the VTA. Specifically, VTA-targeting glutamatergic neurons from the mPFC and the PPN, but not the LHb, express the CB1 receptor. Our observations of pathway-specific modulation by oxytocin demonstrate a mechanism for selective gating of information flowing into VTA reward systems. For OxtR-expressing VTA DA neurons, one possibility is that CB1 receptor positive and negative glutamatergic signals converge onto single DA neurons, leading to selective filtering of inputs in the presence of oxytocin. However, an alternative hypothesis consistent with our observations is that some VTA DA neurons receive exclusively CB1 receptor positive or negative glutamatergic inputs. In addition, not all VTA DA neurons express oxytocin receptors. The details of topographic mapping between glutamatergic inputs and VTA DA neurons, with regard to their oxytocin responsiveness, remain to be characterized in future studies.

Regardless of where the pathway-selectivity arises, evidence in support of pathway-specific oxytocin modulation of synaptic transmission to VTA DA neurons is likely to have functional consequences for information processing and behavior. Distinct glutamatergic inputs to VTA DA systems are associated with varied behavioral output. For example, the lateral habenula acts as a source of negative reward signals for DA neurons (*Lammel et al., 2012*; *Matsumoto and Hikosaka, 2007*). Meanwhile, mPFC glutamatergic afferents increase VTA activity and may promote the development of addiction (*Gariano and Groves, 1988*; *Wu et al., 2013*), and pedunculopontine glutamatergic afferents to VTA DA neurons drive motivated behavior (*Yoo et al., 2017*). One possibility that can be explored in future studies is that oxytocin selectively gates inputs of specific valence.

Oxytocin in the VTA has recently been shown to promote socially rewarding behavior (*Hung et al., 2017*). This study also reported a direct facilitation of tonic activity in VTA DA neurons, in response to oxytocin receptor agonist application, consistent with our prior observations (*Xiao et al., 2017*). Hung et al. propose that an increase in excitatory drive to VTA DA neurons, with a change in excitation/inhibition balance, accounts for the enhancement of tonic activity in response to oxytocin. These data appear to contradict our observations of oxytocin-evoked endocannabinoid-mediated dampening of EPSC amplitudes, but differences in neuronal identity and recording conditions likely account for the observed discrepancies. First, Hung et al. targeted VTA DA neurons retrograde-labeled from medial nucleus accumbens (NAc), while we recorded non-selectively targeted ventromedial DA neurons. This difference in selection is important because DA neurons projecting to distinct regions of NAc receive different glutamatergic inputs (*Beier et al., 2015*). Medial NAc-projecting VTA DA neurons are innervated by glutamatergic axons from lateral hypothalamus, lateral habenula and dorsal raphe nuclei (*Beier et al., 2015*), where we do not observe extensive presence of CB1Rs in glutamatergic neurons. Additionally, Hung et al. include spermine in their recording pipette solution, presumably in order to intracellularly block NMDARs (*Araneda et al., 1999*) for tight pharmacological control important when recording EPSCs and IPSCs from the same neurons. However, spermine can modulate and inhibit the PLC cascade (*Sechi et al., 1978*; *Wojcikiewicz and Fain, 1988*) which is downstream of the OxtR. The inclusion of spermine in the internal solution is expected to emphasize circuit-level effects of oxytocin application or release over cell-autonomous effects that rely on retrograde signaling. Altogether, both studies using distinct targeting and pharmacological recording conditions produce important insights into oxytocin signaling in the VTA, highlighting the complexity of this multifaceted modulation and raising questions for further study.

In the VTA, oxytocin is poised to function as a selective high-pass filter for synaptic transmission in several distinct ways. First, the heterogeneity of CB1 receptor expression in glutamatergic projections means that some inputs are allowed to pass through with no dampening in the presence of oxytocin. This mechanism also operates in the striatum, where cortical glutamatergic inputs, but not thalamostriatal ones, display CB1 receptor-dependent LTD (*Wu et al., 2015*). Second, in our experiments only the initial stimuli in pairs or trains appear to be decreased by oxytocin. Therefore, on a short time-scale repetitively active inputs, presumably carrying behaviorally salient information, are allowed to pass. Moreover, transient exposure to oxytocin is sufficient to lead to relatively long-lasting changes in synaptic transmission. On a longer time-scale, oxytocin signaling does not potentiate the canonical LTD mechanism itself, but it can provide a separate endocannabinoid-dependent, presynaptic component to forms of long-term plasticity that are important for VTA DA neurons. VTA DA LTD has been implicated in addiction, fear learning, satiety, and aversion behaviors (*Labouèbe et al., 2013*; *Liu et al., 2010*; *Pignatelli et al., 2017*; *Thomas and Malenka, 2003*). Therefore, the additive effects of oxytocin to classical LTD increase the range of physiological contexts where neurohypophyseal peptide signaling impacts the VTA.

Altogether, our findings suggest that oxytocin gates inputs into reward systems both spatially and temporally. How would oxytocin-mediated dampening of excitatory synaptic transmission operate in the context of oxytocin-evoked enhancement of tonic activity in VTA DA neurons that we and others have reported (*Hung et al., 2017*; *Tang et al., 2014*; *Xiao et al., 2017*)? In midbrain DA neurons, the activation of $G\alpha_q$-coupled cascades enhances endocannabinoid release (*Gantz and Bean, 2017*). OxtR signaling canonically occurs through $G\alpha_q$ cascades; mechanistically, the corresponding increases in calcium concentration may provide the intracellular calcium required for endocannabinoid synthesis and release. Conceptually, an increase in the excitability of VTA DA neurons, together with a decoupling from a subset of inputs, may leave these neurons primed for action in response to

the inputs that show persistent activity or are not dampened by endocannabinoid retrograde actions.

This new evidence for oxytocinergic control of synaptic transmission in the VTA has several clinical implications, given the involvement of midbrain DA regions in reward-based behavior (*Howe and Dombeck, 2016*; *Russo and Nestler, 2013*; *Schultz, 1998*), drug abuse and addiction (*Jones and Bonci, 2005*; *Kauer and Malenka, 2007*; *Niehaus et al., 2009*; *Stelly et al., 2016*), as well as neuro-developmental and neurodegenerative disorders (*Dichter et al., 2012*). The existence of this modulatory system opens new possibilities for indirectly controlling endocannabinoid or dopaminergic signaling by leveraging interacting neuromodulators like oxytocin. Endocannabinoid signaling is suggested to be neuroprotective across multiple brain regions, including the hippocampus and midbrain dopamine regions (*Melis and Pistis, 2007*; *Xu and Chen, 2015*). Pharmacologically enhancing endocannabinoid-mediated striatal plasticity suffices to rescue motor deficits observed in rodent models of Parkinson's disease (*Kreitzer and Malenka, 2007*). Given the vast literature on the neuroprotective properties of oxytocin (*Ceanga et al., 2010*; *Kaneko et al., 2016*; *Tyzio et al., 2006*) and the broad expression of oxytocin receptors throughout the vertebrate brain (*Mitre et al., 2016*), the therapeutic potential of developing oxytocin receptor-targeting adjunctive pharmacological agents could be considerable. Because endocannabinoids in midbrain dopamine systems are linked to the processing of socially and non-socially rewarding stimuli, including drugs of abuse, oxytocinergic control over endocannabinoid signaling establishes new research questions as well as possibilities for therapeutic interventions.

# Materials and methods

## Key resources table

| Reagent type (species) or resource | Designation | Source or reference | Identifiers | Additional information |
|---|---|---|---|---|
| strain, strain background (Mouse) | C57BL/6 | Charles River | Cat#000664; RRID:IMSR_JAX:000664 | |
| strain, strain background (Mouse) | B6.SJL-Slc6a3 $^{tm1.1(cre)Bkmn}$/J | Jackson Laboratory | Cat#006660; RRID: IMSR_JAX:006660 | |
| strain, strain background (Mouse) | B6.129S-Oxt$^{tm1.1(cre)Dolsn}$/J | Jackson Laboratory | Cat#024234; RRID:IMSR_JAX:024234 | |
| strain, strain background (Mouse) | Slc17a6 $^{tm2(cre)Lowl}$/J | Jackson Laboratory | Cat#016963; RRID:IMSR_JAX:016963 | |
| strain, strain background (Mouse) | B6.Cg-Gt(ROSA) 26Sor$^{tm14(CAG-tdTomato)Hze}$/J | Jackson Laboratory | Cat#007914; RRID:IMSR_JAX:007914 | |
| strain, strain background (Mouse) | B6.129S-Gt(ROSA)26Sor$^{tm32}$ $^{(CAG-COP4*H134R/EYFP)Hze}$/J | Jackson Laboratory | Cat#012569; RRID:IMSR_JAX:012569 | |
| antibody | Rabbit anti-tyrosine hydroxylase | Millipore | Cat#AB152; RRID:AB_390204 | concentration: 1:1000 |
| antibody | Rabbit anti-oxytocin receptor (OxtR-2) | NA | NA | Gift from R. Froemke (*Marlin et al., 2015*; *Mitre et al., 2016*) |
| recombinant DNA reagent | AAV9-EF1a-DIO-hChR2 (H134R)-mCherry | UPenn viral core | CS0543-3CS | |
| commercial assay or kit | RNAscope Fluorescence Multiplex Assay | ACDBio | | |
| commercial assay or kit | RNAscope Probe- Mm-Cnr1 | ACDBio | Cat#420721-C1 | |
| commercial assay or kit | RNAscope Probe- Mm-Th | ACDBio | Cat#317621-C2 | |
| commercial assay or kit | RNAscope Probe- Mm-Slc17a6 | ACDBio | Cat#319171-C2 | |
| commercial assay or kit | RNAscope Probe- Mm-Slc17a7 | ACDBio | Cat#416631-C2 | |
| commercial assay or kit | RNAscope Probe- Mm-Oxtr | ACDBio | Cat#412171-C1 | |
| chemical compound, drug | Oxytocin | Tocris | Cat#1910; CAS 50-56-6 | |
| chemical compound, drug | L-368,899 hydrochloride | Tocris | Cat#2641; CAS 160312-62-9 | |

*Continued on next page*

*Continued*

| Reagent type (species) or resource | Designation | Source or reference | Identifiers | Additional information |
|---|---|---|---|---|
| chemical compound, drug | SR 49059 | Tocris | Cat#2310; CAS 150375-75-0 | |
| chemical compound, drug | NBQX disodium salt | Tocris | Cat#1044; CAS 479347-86-9 | |
| chemical compound, drug | (RS)-CPP | Tocris | Cat#0173; CAS 100828-16-8 | |
| chemical compound, drug | SR95531 hydrobromide | Tocris | Cat#1262; CAS 104104-50-9 | |
| chemical compound, drug | U73122 | Tocris | Cat#1268; CAS 112648-68-7 | |
| chemical compound, drug | CP945598 hydrochloride | Tocris | Cat#4236; CAS 686347-12-6 | |
| chemical compound, drug | WIN55212-2 | Tocris | Cat#1038; CAS 131543-23-2 | |
| chemical compound, drug | AM251 | Tocris | Cat#1117; CAS 183232-66-8 | |
| chemical compound, drug | orlistat (THL) | Tocris | Cat#3540; CAS 96829-58-2 | |
| chemical compound, drug | MNI-L-glutamate | Tocris | Cat#1490; CAS 295325-62-1 | |
| chemical compound, drug | Alexa Fluor 488 hydrazide | Thermo Fisher Scientific | Cat#A10436 | |
| chemical compound, drug | Alexa Fluor 594 hydrazide | Thermo Fisher Scientific | Cat#A10438 | |
| chemical compound, drug | Green retrobeads (GRB) | Lumafluor Inc. | Cat#G180 | |
| software, algorithm | GraphPad Prizm 5 | GraphPad | RRID:SCR_002798 | |
| software, algorithm | FIJI | Schindelin et al., 2012 | http://fiji.sc/; RRID:SCR_002285 | |
| software, algorithm | MATLAB | MathWorks | RRID:SCR_001622 | |
| software, algorithm | Igor Pro | Wavemetrics | RRID:SCR_000325 | |
| software, algorithm | SPSS | IBM | RRID:SCR_002865 | |

## Mouse strains and genotyping

Animals were handled according to protocols approved by the Northwestern University Animal Care and Use Committee. Weanling and young adult mice (postnatal days 25–40) of both sexes were used in this study. C57BL/6 mice used for breeding were acquired from Charles River (Wilmington, MA); other mouse lines were acquired from the Jackson Laboratory (Bell Harbor, ME). B6.SJL-Slc6a3 $^{tm1.1(cre)Bkmn}$/J mice (*Slc6a3*$^{i-Cre}$), which express Cre recombinase under the control of the dopamine transporter promoter (*Bäckman et al., 2006*), were used to identify dopaminergic neurons via reporter crosses. B6.129S-*Oxt* $^{tm1.1(cre)Dolsn}$/J mice (*Oxt*$^{i-Cre}$, #024234), which express the enzyme Cre recombinase under control of the oxytocin promoter (*Shah et al., 2014*), were used to target oxytocinergic neurons in PVN. *Slc17a6*$^{tm2(cre)Lowl}$/J mice (*Slc17a6*$^{i-Cre}$, # 016963), which express Cre recombinase under the control of vesicular glutamate transporter 2 promoter (*Vong et al., 2011*), were used to target glutamatergic neurons. *Slc6a3*$^{i-Cre}$ mice were crossed to a floxed tdTomato reporter strain (Ai14, Jackson Lab, #007914), or, for a small subset of experiments, a floxed ChR2-eYFP strain (Ai32, Jackson lab, # 012569) (*Madisen et al., 2010*). Mice heterozygous for Cre were used for experiments; genotyping followed standard procedures available on the Jackson Lab website.

## Stereotaxic intracranial injections

P23-25 mice were anesthetized with 1.5–2% isofluorane, received ketoprofen for analgesia, and were placed on a small animal stereotaxic frame (David Kopf Instruments, Tujunga, CA). Green retrobeads (Lumafluor, Naples, FL) were delivered unilaterally into the VTA (2.7 mm posterior to bregma, 0.5 mm lateral, and 4.5 mm below the pia) through a pulled glass pipette at a rate of 50 nl/minute for a total of 100 nl using an UltraMicroPump (World Precision Instruments, Sarasota, FL). AAV9-EF1a-DIO-hChR2(H134R)-mCherry rAAV (1.24 × 10$^{13}$ GC/ml) was unilaterally injected into mPFC (2.4 mm anterior to bregma, 0.4 mm lateral, and 2.4 mm below the pia), LHb (1.0 mm posterior to bregma, 0.3 mm lateral, and 2.6 mm below the pia), or PPN (4.7 mm posterior to bregma, 1.3 mm lateral, and 3.7 mm below the pia) at a rate of 100 nl/minute for a total of 150 nl. For targeting oxytocinergic neurons, the same virus was injected into the PVN (1.0 mm posterior to bregma, 0.3 mm lateral, 4.5 mm and 4.7 mm below the pia) at a rate of 100 nl/minute for a total of 1000 nl. Injection coordinates for green retrobeads in the VTA: 2.7 mm posterior to bregma, 0.5 mm lateral, and 4.5

mm below the pia. The pipette was held at the injection location for 15–20 min after retrobead injection and 10 min after virus injection. Coordinates were slightly adjusted for mouse age and size. Mice recovered for 7–9 days following retrograde tracer injections, and 4–5 weeks after virus injection.

## Tissue processing, immunohistochemistry and imaging

To confirm neuronal identity as dopaminergic cells in all electrophysiology experiments combined with optogenetics, slices were fixed in 4% paraformaldehyde (PFA) overnight after recording, washed in 0.1 M phosphate buffed saline (PBS), and processed for immunostaining against tyrosine hydroxylase. Sections were pretreated in 0.2% Triton-X100 for an hour at RT, and then incubated for 24 hr at 4°C with primary antibody solution in PBS with 0.2% Triton-X100 (rabbit anti-tyrosine hydroxylase, 1:1000; AB152, Abcam, Cambridge, UK). Tissue was rinsed in PBS, reacted with secondary antibody for 2 hr at RT (goat anti-rabbit Alexa 647, 1:500, Life Technologies, Carlsbad, CA), rinsed again, then mounted onto Superfrost Plus slides (ThermoFisher Scientific, Waltham, MA), dried and coverslipped under glycerol:TBS (3:1) with Hoechst 33342 (1:1000; ThermoFisher Scientific). OxtR-2 antibody staining (a generous gift of R. Froemke) was conducted following previously published procedures (*Mitre et al., 2016*). Mice were deeply anaesthetized with isoflurane and transcardially perfused with 4% PFA. Brains were post-fixed for 2 hr, and transferred to 30% sucrose solution in PBS and stored at 4°C overnight. Then, brains were embedded in Tissue-Tek O.C.T. compound (VWR), stored overnight at −80°C, and sectioned on a cryostat at 18 μm thickness. Sections were rinsed in PBS, blocked for 2 hr in PBS with 0.2% Triton X-100% and 5% donkey serum, and incubated with OxtR-2 antibody serum at 1:250 dilution. Following a 2 day-long incubation at 4°C in a humidified chamber, sections were rinsed three times in PBS and incubated for 2 hr in Alexa Fluor 647-conjugated goat anti-rabbit antibody (Thermo Fisher Scientific, 1:500). Whole sections were imaged with an Olympus VS120 slide scanning microscope (Olympus Scientific Solutions Americas, Waltham, MA). Confocal images were acquired with a Leica SP5 confocal microscope (Leica Microsystems). Depth-matched z-stacks of 1 μm-thick optical sections were analyzed in ImageJ (FIJI) (*Schindelin et al., 2012*).

## Quantitative fluorescence in situ hybridization (FISH)

FISH labeling and analyses were conducted according to previously published procedures (*Xiao et al., 2017*). Briefly, brains were quickly removed from deeply anesthetized mice and frozen in tissue-freezing medium prior to storage at −80°C, sectioned at 20 μm (Leica CM1850), adhered to Superfrost Plus slides, and frozen. Samples were fixed with 4% PFA in 0.1 M PBS at 4°C, processed according to RNAscope Fluorescent Multiplex Assay manual for fresh frozen tissue (Advanced Cell Diagnostics, Newark, CA), and coverslipped using ProLong Gold antifade reagent with DAPI (Molecular Probes). The following probes were used: *Cnr1* and *Oxtr* in channel 1, with tyrosine hydroxylase (*Th*), *Slc17a6 (Vglut2)*, or *Slc17a7 (Vglut1)* in channel 2. Sections were subsequently imaged on a Leica SP5 confocal microscope in three channels with a 40x or 100x objective lens, with 1 μm between adjacent z-sections. Probe omission negative controls were carried out for every reaction.

FISH images were analyzed as previously (*Xiao et al., 2017*) with a MATLAB script utilizing imreadBF for file loading and a modified version of Fast 2D peak finder. Three adjacent z-stack slices were combined, for a total of ~3 μm of tissue. In general, combining between 2 and 3 μm was optimal to ensure that differences in subcellular localization of RNA transcripts do not lead to missed co-localization, while minimizing false positive co-localization driven by signal from other cells. All channels were thresholded for intensity to remove background signal. Watershed segmentation of the image was performed using *Th*, *Slc17a6*, or *Slc17a7* channel information to localize somata. Puncta of FISH molecules were counted within established cell boundaries. Whether a cell was considered positive for a given marker was determined by setting transcript-dependent thresholds for the number of puncta, but data for all imaged cells are shown throughout the study. This threshold was set by comparing manual counts of cells to histograms of puncta per cell for several images. The established puncta number threshold was used for all remaining images of a given channel/probe combination.

## Acute slice preparation and electrophysiology

Brain slice preparation was adapted from previously published procedures (*Xiao et al., 2017*). Briefly, animals were deeply anesthetized by isoflurane, followed by a transcardial perfusion with ice-cold, oxygenated artificial cerebrospinal fluid (ACSF) containing (in mM) 127 NaCl, 2.5 KCl, 25 NaHCO$_3$, 1.25 NaH$_2$PO$_4$, 2.0 CaCl$_2$, 1.0 MgCl$_2$, and 25 Glucose (osmolarity ~310 mOsm/L). After perfusion, the brain was removed and immersed in ice-cold ACSF. Tissue was blocked and transferred to a slicing chamber containing ice-cold ACSF, supported by a block of 4% agar. Horizontal slices of 250 μm thickness were cut on a Leica VT1000s in ventral-dorsal direction and transferred into a holding chamber with ACSF, equilibrated with 95%O$_2$/5%CO$_2$. Slices were incubated at 34 ˚C for ~30 min prior to electrophysiological recording.

Slices were transferred to a recording chamber perfused with oxygenated ACSF at a flow rate of 2–4 ml/min at room temperature. Whole-cell recordings were obtained from neurons visualized under infrared DODT contrast video microscopy using patch pipettes of ~2–5 MΩ resistance. For electrical stimulation experiments, VTA dopamine neurons were identified based on tdTomato or eYFP signal in *Slc6a3* [i-Cre]; Ai14 or *Slc6a3*[i-Cre]; Ai32 mice. For optogenetic experiments, dopamine neurons were identified on the basis of the combination of their electrophysiological and morphological properties, and further validated with *post hoc* TH immunolabeling (*Xiao et al., 2017*). Recording electrodes contained the following (in mM): 120 CsMeSO$_4$, 15 CsCl, 10 HEPES, 10 Na-phosphocreatine, 2 MgATP, 0.3 NaGTP, 10 QX314, and 1 EGTA (pH 7.2-7.3, ~295 mOsm/L). For optogenetic and two-photon uncaging experiments, Alexa Fluor 488 dye (20 μM) was added to the internal solution. SR95531 (10 μM) was added to ACSF for recording electrical stimulation-evoked EPSCs, light stimulation-evoked EPSCs, and spontaneous EPSCs, all acquired at a holding potential of −70 mV. Recordings were made using 700B amplifiers (Axon Instruments, Union City, CA); data were sampled at 10 kHz and filtered at 4 kHz with a MATLAB-based acquisition script (MathWorks, Natick, MA). Series resistance and input resistance were monitored using a 100 ms, 5 mV hyperpolarizing pulse at every sweep, and experiments were started after series resistance had stabilized (~20 MΩ, uncompensated).

For electrical stimulation experiments, a concentric bipolar micro electrode (CBAPB75, FHC, Inc) was placed approximately 100 μm away from the recording electrode and 80 μs electrical pulses applied at intervals of 30 s were used to evoke EPSCs. The amplitudes of EPSCs were calculated by taking a 1 ms window around the peak of the EPSC and comparing this to a 1 s window immediately prior to the onset of the electrical stimulation artefact. Paired stimuli were delivered using inter-stimulus intervals of 50, 80 and 100 ms, and the paired-pulse ratio (PPR) was defined as the ratio between the amplitudes of the second and the first EPSCs. EPSC decay time was calculated based on a single exponential fit and reported as the time constant, averaged for all recorded EPSCs within each neuron in baseline condition and following oxytocin application. Spontaneous EPSCs (sEPSCs) were pharmacologically isolated in the presence of GABA(A)R antagonist SR 95531 (10 μM).

A 5 min, 10 Hz repetitive electrical stimulation was used to induce LTD of EPSCs in VTA DA neurons (*Pan et al., 2008*). In a subset of experiments LTD was induced using 1 s long periods of 100 Hz stimulation, repeated 4 times with 10 second-long inter-train intervals (*Kreitzer and Malenka, 2005*). Paired stimuli delivered at 50 ms inter-stimulus interval and 30 second-long inter-train intervals were used to evoke EPSCs before and after inducing LTD. The amplitude of first EPSC in every train was used to quantify EPSC amplitude change, and PPR was defined as the ratio between the amplitudes of the second and the first EPSCs. For quantifying the magnitude of LTD, EPSCs were normalized based on 5 min baseline recordings. LTD magnitude reflects the average normalized EPSC amplitude during a 10 min-long period starting 10 min after the end of induction period.

To activate ChR2-expressing fibers of glutamatergic neurons in the VTA, 2 ms-long light pulses (470 nm, ~5 mW) at intervals of 30 s were delivered at the recording site using whole-field illumination through a 40X water-immersion objective (Olympus, Tokyo, Japan) with a PE300 CoolLED illumination system (CoolLED Ltd., Andover, UK). The amplitudes of optical evoked EPSCs were calculated by taking a 1 ms window around the peak of the EPSC and comparing this to a 1 s window prior to the onset of the light stimulation. To activate ChR2-expressing fibers of oxytocinergic neurons in the VTA, 10 ms-long light pulses (470 nm, 20 Hz for 30 s, ~5 mW) were delivered at the recording site using whole-field illumination. Electrically evoked EPSCs were recorded 60 s, 30 s,

and 0 s before light stimulation, and 0 s and 30 s after light stimulation. Three to four consecutive responses at 6 min-long intervals were acquired for each neuron.

## Two-photon laser-scanning microscopy with glutamate uncaging

$Slc6a3^{i-Cre}$; tdTomato mice were used for all two-photon uncaging and imaging experiments. We recorded dopamine neurons in the VTA using cesium-based internal solution containing 20 µM Alexa Fluor 488. After a 10–15 min long wash-in period, cell morphology was visualized using Alexa Fluor signal. Two mode-locked Ti:Sapphire lasers (Mai Tai eHP DS, Newport) were used for imaging and uncaging at the wavelengths of 910 nm and 720 nm, respectively. The beam of the laser was directed by a two-dimensional galvanometer scanning mirror system (HSA Galvo 8315K, Cambridge Technology). Fluorescence emission was collected by two PMTs above and below the sample (H10770P, Hamamatsu) after passing through a dichroic beamsplitter (FF670-SDi01−26 × 38, Semrock) and a bandpass filter (FF02- 520/28, Semrock). 2.5 mM MNI-L-glutamate (Tocris) was perfused in the recirculating bath and two 0.5 ms long laser pulses at 50 ms ISI were delivered to a target spot near a dendritic spine to photoactivate glutamate. We used a version of Scanimage to control scanning parameters and image acquisition (*Kozorovitskiy et al., 2015*; *Pologruto et al., 2003*). Laser intensity was controlled by a Pockels cell and laser power at the sample plane was 10–15 mW. After selecting a dendritic spine for uncaging, several spots around the spine were sampled for maximal uncaging-evoked EPSC (uEPSC) response. Location of the maximal response for a given dendritic spine was selected for the experiment. At least ten consecutive AMPAR-uEPSCs were acquired at 1 second-long intervals from each dendritic spine.

## Pharmacology

Pharmacological agents were acquired from Tocris (Bristol, UK) or Sigma-Aldrich (St. Louis, MO). Drugs were applied by bath perfusion: oxytocin (10 nM - 1 µM), L368,899 hydrochloride (10 µM), U73122 (10 µM), SR49059 (10 µM), CP945598 hydrochloride (10 µM), WIN55212-2 (5 µM), AM251 (5 µM), orlistat (THL, 5 µM)), MNI-L-glutamate (2.5 mM), SR95531 hydrobromide (10 µM), NBQX (10 µM), and CPP (10 µM).

## Quantification and statistical analysis

Offline analyses of electrophysiology were performed using MATLAB (Mathworks, Natick, MA) and IgorPro (Wavemetrics, Portland, OR). Whenever possible, data were analyzed blind to condition. For sample sizes, both the number of neurons analyzed and the number of animals are provided. Sex and age were balanced across groups. Statistical analyses were carried out using GraphPad Prizm 5 software (GraphPad, LaJolla, CA) and SPSS (IBM, New York, NY). Group data are expressed as group means ±SEM. Only non-parametric statistical tests are used throughout the study. For two-group comparisons, statistical significance was determined by two-tailed Wilcoxon matched-pairs signed rank test or Mann-Whitney test. For multiple group comparisons, Friedman's 2-Way ANOVA by ranks test and Kruskal-Wallis test with Dunn's Multiple Comparison *post hoc* test were used. $p<0.05$ was considered statistically significant.

## Acknowledgements

We thank Lindsey Butler for genotyping and colony management, as well as all Kozorovitskiy lab members for comments. We thank the Northwestern University Biological Imaging Facility and Dr. Tiffany Schmidt for confocal microscope access, and the Transgenic and Targeted Mutagenesis Laboratory for cryorecovery of mouse lines. This work was supported by the Beckman Young Investigator Award, William and Bernice E Bumpus Young Innovator Award, Rita Allen Foundation Scholar Award, Sloan Research Fellowship, and Searle Scholar Award (all YK). MFP was supported by an Arnold O. Beckman Postdoctoral Fellowship.

# Additional information

## Funding

| Funder | Grant reference number | Author |
| --- | --- | --- |
| Arnold and Mabel Beckman Foundation | Arnold O. Beckman Postdoctoral Fellowship | Michael F Priest |
| Arnold and Mabel Beckman Foundation | BYI Award | Yevgenia Kozorovitskiy |
| William and Bernice E Bumpus Foundation | Young Innovator Award | Yevgenia Kozorovitskiy |
| Rita Allen Foundation | Scholar Award | Yevgenia Kozorovitskiy |
| Alfred P. Sloan Foundation | Research Fellowship | Yevgenia Kozorovitskiy |
| Kinship Foundation | Searle Scholar Award | Yevgenia Kozorovitskiy |

The funders had no role in study design, data collection and interpretation, or the decision to submit the work for publication.

## Author contributions

Lei Xiao, Conceptualization, Resources, Data curation, Formal analysis, Supervision, Funding acquisition, Validation, Investigation, Visualization, Methodology, Writing—original draft, Project administration, Writing—review and editing; Michael F Priest, Conceptualization, Resources, Data curation, Software, Formal analysis, Validation, Investigation, Visualization, Methodology, Writing—original draft, Project administration, Writing—review and editing; Yevgenia Kozorovitskiy, Software, Methodology, Writing—original draft, Writing—review and editing

## Author ORCIDs

Lei Xiao (iD) http://orcid.org/0000-0002-1640-9690
Michael F Priest (iD) http://orcid.org/0000-0002-4306-6937
Yevgenia Kozorovitskiy (iD) http://orcid.org/0000-0002-3710-1484

## Ethics

Animal experimentation: Animals were handled according to protocols approved by the Northwestern University Animal Care and Use Committee (IS00002086 and IS00000707).

## Decision letter and Author response

Decision letter https://doi.org/10.7554/eLife.33892.026
Author response https://doi.org/10.7554/eLife.33892.027

# Additional files

## Supplementary files

• Transparent reporting form
DOI: https://doi.org/10.7554/eLife.33892.024

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
