## [Decision Letter]

Thank you for submitting your article "Oxytocin as a spatiotemporal gate for VTA dopamine neuron excitatory synaptic transmission" for consideration by *eLife*. Your article has been reviewed by three peer reviewers, and the evaluation has been overseen by a Reviewing Editor and Gary Westbrook as the Senior Editor. The following individuals involved in review of your submission have agreed to reveal their identity: Valéry Matarazzo (Reviewer #1); Stephanie C Gantz (Reviewer #2).

The reviewers have discussed the reviews with one another and the Reviewing Editor has drafted this decision to help you prepare a revised submission.

Summary:

The study by Xiao et al., reports a pathway-specific suppression of glutamatergic synaptic transmission by oxytocin via retrograde endocannabinoid-dependent suppression of glutamate release. The data are well-organized and clearly illustrated. A strength to the study is the use of optogenetic activation of oxytocinergic axons from the PVN to the VTA to endogenously release oxytocin.

Essential revisions:

These four major concerns, that likely will require additional experiments must be addressed.

1) Are the OXT neurons synaptically connected to VTA DA neurons? If so, what is the contribution of the glutamate released from OXT neurons on EPSCs?

Although results from Figure 2G-I suggest that oxytocin does not modulate post-synaptic properties, could optical activation of oxytocin-positive axons in VTA combined with glutamate uncaging-evoked EPSCs (or other approach) be performed to address a post-synaptic effect of OXT fibers through glutamate released by OXT axons terminals?

2) The claim that oxytocin is acting on oxytocin receptors on dopamine neurons is not well-substantiated. All of the manipulations to disrupt OxtR signaling (L368,899, U73122, and THL) would also disrupt signaling in all other cell populations. Oxytocin receptor signaling in dopamine neurons could be demonstrated by disrupting signaling in the recorded neuron (e.g. intracellular U73122, THL, or GDPβ-S, or selective deletion of OxtR in dopamine neurons). In light of their previous paper demonstrating oxytocin enhanced tonic firing of VTA dopamine neurons (an effect that did not require glutamatergic or GABAergic signaling), it is not clear as to why no direct effects of oxytocin on dopamine neurons were reported in this study. This becomes very important in the interpretation of the pathway specific modulation presented in Figure 5 and Figure 6—figure supplement 1. It is not clear that the dopamine neurons that receive inputs from LHb have oxytocin receptors.

3) It is unclear how oxytocin is being released with optical activation of PVN fibers but not during repetitive electrical stimulation used to induce LTD. The claim that "oxytocin regulates short and long-term synaptic plasticity" is better supported by demonstrating that the magnitude of LTD is reduced/reversed by OxtR antagonism or that optical activation of oxytocin fibers in the VTA produces short-term or LTD. Alternatively, is it expected that dopamine neurons would be exposed to oxytocin for 30 minutes under physiological conditions? If so, this information should be included to help the reader understand the physiological relevance.

4) The finding that oxt 'enhances long-term depression' is somewhat confusing (Figure 6D-G). What the authors seem to have uncovered is an additive effect of endocannabinoid release (driven by oxt and stimulation), with a standard postsynaptic form of LTD. It is not sure the authors would agree that it's not the case that oxt has somehow potentiated the LTD mechanism itself. This could be clarified in a couple ways: 1) assuming that this form of LTD is NMDAR-dependent, conduct the same experiment in the presence of APV. There should be some residual "LTD" that is due to oxt+stim induced endocannabinoid release. 2) show that oxt-induced endocannabinoid release causes long-term suppression of EPSCs. For example, all of Figures1-3 show experiments ending after 10 min of drug application, rather than measuring EPSCs during a 'washout' period. Showing that oxt-induced depression lasts longer would be consistent with the "LTD"-like mechanisms implied in Figure 6.

---

## [Author Response]

Essential revisions:These four major concerns, that likely will require additional experiments must be addressed.1) Are the OXT neurons synaptically connected to VTA DA neurons? If so, what is the contribution of the glutamate released from OXT neurons on EPSCs?Although results from Figure 2G-I suggest that oxytocin does not modulate post-synaptic properties, could optical activation of oxytocin-positive axons in VTA combined with glutamate uncaging-evoked EPSCs (or other approach) be performed to address a post-synaptic effect of OXT fibers through glutamate released by OXT axons terminals?

We agree with the reviewers that the possibility of Oxt neurons releasing glutamate in the VTA should be explored, since (1) this co-release has been reported in the parabrachial nucleus (Ryan et al., 2017) and in brainstem vagal neurons (Pinol et al., 2014); and (2) a positive finding would substantially alter interpretation of the major results of this study. To evaluate potential co-release, we expressed ChR2 in PVN oxytocin neurons and recorded VTA DA neurons using whole-cell voltage clamp. We used short single light pulses to optically stimulate Oxt fibers, as has been done in the manuscripts referenced above to evoke > 50 pA amplitude EPSCs. In contrast to parabrachial and vagal neurons, we saw no fast responses in VTA DA cells (light-evoked response change -0.556 ± 0.285 pA, p = 0.3034, Wilcoxon matched-pairs signed rank test; 16 neurons, 5 mice). We conclude that PVN Oxt fibers do not co-release glutamate in the VTA and have modified the Results section accordingly.

2) The claim that oxytocin is acting on oxytocin receptors on dopamine neurons is not well-substantiated. All of the manipulations to disrupt OxtR signaling (L368,899, U73122, and THL) would also disrupt signaling in all other cell populations. Oxytocin receptor signaling in dopamine neurons could be demonstrated by disrupting signaling in the recorded neuron (e.g. intracellular U73122, THL, or GDPβ-S, or selective deletion of OxtR in dopamine neurons).

To evaluate cell-autonomous function of the PLC/endocannabinoid pathway in mediating Oxt modulation of VTA DA EPSCs, we measured Oxt modulation of electrically evoked EPSCs in VTA DA neurons, while using intracellular inhibitors of PLC or of 2-AG synthesis (U73122 or THL respectively) in the internal solution. As we had previously reported for bath application of these agents, cell-delimited manipulations were sufficient to abolish the effects of Oxt on EPSCs (Figure 3E-G). These data support the argument that Oxt acts directly on VTA DA neurons to decrease EPSC amplitude.

In light of their previous paper demonstrating oxytocin enhanced tonic firing of VTA dopamine neurons (an effect that did not require glutamatergic or GABAergic signaling), it is not clear as to why no direct effects of oxytocin on dopamine neurons were reported in this study. This becomes very important in the interpretation of the pathway specific modulation presented in Figure 5 and Figure 6—figure supplement 1. It is not clear that the dopamine neurons that receive inputs from LHb have oxytocin receptors.

To clarify, in this study we also observed a direct effect of Oxt on excitability of VTA DA neurons, evidenced by a change in holding current in response to optogenetic stimulation of Oxt fibers for voltage-clamped VTA DA neurons (2.955 ± 0.871 pA, p < 0.01, Wilcoxon Signed Rank Test, 15 neurons, 8 mice). As we have reported previously (Figure S4 in Xiao et al., 2017), a <5 pA current injection into VTA DA neurons suffices to induce a change in tonic firing rate that is comparable to tonic activity changes in response to Oxt, recorded in current-clamp mode. We have carried out this analysis specifically for optogenetic stimulation of Oxt fibers, rather than bath application experiments, because of the tight temporal control afforded by this paradigm. Thus, our new data are consistent with (Xiao et al., 2017), but add an important element into the circuit-level schematic for Oxt actions in the VTA. We have added this new analysis to the Results section.

As correctly pointed out by the reviewers, we cannot completely exclude the possibility that VTA DA neurons receiving LHb innervation do not express OxtRs. On one hand, we targeted VTA DA neurons for all pathway manipulation experiments in the same manner, within ventromedial VTA where OxtRs are enriched (new panels Figure 1—figure supplement 1A-C). On the other hand, since less than half of all VTA DA neurons show LHb axon optical stimulation-evoked EPSCs, it is possible that those cells selectively lack OxtRs. Regardless, the overall conclusion about pathway-selective Oxt effects on glutamatergic signaling still stands. We have expanded the Results and Discussion sections to acknowledge this important alternative interpretation of pathway-selective modulation of EPSCs.

3) It is unclear how oxytocin is being released with optical activation of PVN fibers but not during repetitive electrical stimulation used to induce LTD. The claim that "oxytocin regulates short and long-term synaptic plasticity" is better supported by demonstrating that the magnitude of LTD is reduced/reversed by OxtR antagonism or that optical activation of oxytocin fibers in the VTA produces short-term or LTD. Alternatively, is it expected that dopamine neurons would be exposed to oxytocin for 30 minutes under physiological conditions? If so, this information should be included to help the reader understand the physiological relevance.

Given our data on dose-dependent effects of Oxt on EPSCs (Figure 1F), it is plausible that repetitive electrical stimulation used to induce LTD (10Hz or 100Hz in two different protocols) causes some Oxt release that elicits submaximal effects, allowing for further EPSC suppression by Oxt that we do observe. If any electrical stimulus used to induce LTD elicited substantial Oxt release that occupied all available receptors, then Oxt perfusion should have no additive effect to LTD magnitude (in contrast to new Figure 4D-G). In addition, we expect little electrically evoked Oxt release, because OxtR antagonist L368,899 blocks Oxt effects on LTD, bringing it to similar magnitude as controls (new Figure 4G). If substantial Oxt tone were driven by LTD induction alone, we would expect the OxtR antagonist (grey dots in Figure 4G) to weaken the magnitude of LTD beyond control values, which we did not observe. We have expanded the Results section accordingly.

Under physiological conditions, high sustained peripheral concentrations of oxytocin are common, especially under conditions of parturition and lactation induction (e.g., ~30 min, Higuchi et al., 1985, J Endocrinol). It is not known whether central elevations of Oxt follow comparable timing, but we do find that transient exposure to Oxt causes EPSC amplitude and PPR changes that outlast the exposure time (new optogenetic experiments in Figure 1). Thus, the EPSC modulation by Oxt appears to be relatively long-lasting, likely due to its underlying mechanism involving endocannabinoid release. This timing stands in contrast to the transient modulation of tonic activity we have reported previously (e.g., Figures 4-5, In Xiao et al., 2017). Thus, a transient Oxt effect on VTA DA neurons can lead to a long-lasting change in synaptic transmission, as has been shown in other systems modulated by Oxt (Dolen et al., 2013; Mitre et al., 2016).

4) The finding that oxt 'enhances long-term depression' is somewhat confusing (Figure 6D-G). What the authors seem to have uncovered is an additive effect of endocannabinoid release (driven by oxt and stimulation), with a standard postsynaptic form of LTD. It is not sure the authors would agree that it's not the case that oxt has somehow potentiated the LTD mechanism itself. This could be clarified in a couple ways: 1) assuming that this form of LTD is NMDAR-dependent, conduct the same experiment in the presence of APV. There should be some residual "LTD" that is due to oxt+stim induced endocannabinoid release. 2) show that oxt-induced endocannabinoid release causes long-term suppression of EPSCs. For example, all of Figures1-3 show experiments ending after 10 min of drug application, rather than measuring EPSCs during a 'washout' period. Showing that oxt-induced depression lasts longer would be consistent with the "LTD"-like mechanisms implied in Figure 6.

We thank the reviewers for the opportunity to clarify our thoughts and observations related to Oxt effect on LTD in VTA DA neurons. Indeed, we find an “additive” pre-synaptic modulation effect of Oxt driven by endocannabinoid release, which increases the magnitude of canonical LTD. The reason to evaluate the possibility that Oxt can provide this additive component to LTD is because this form of plasticity is critically important in VTA DA neurons, and it has been implicated in addiction, fear learning, satiety, and aversion behaviors (e.g., Thomas and Malenka, 2003; Liu et al., 2010;Pignatelli et al., 2017; Labouebe et al., 2013). The observation that Oxt effects on EPSCs can provide an additive component to classical VTA DA neuron LTD increases the range of physiological contexts where Oxt signaling can be important in the VTA. We moved the data describing LTD plasticity experiments to Figure 4to improve the overall flow of the manuscript. Our new data demonstrate that even transient exposure to Oxt is sufficient to lead to a lasting decrease in EPSCs (Figure 1), which would be expected to account for the observations in LTD experiments. The PPR increase after LTD induction (Figure 4—figure supplement 1) also supports the presynaptic effect of Oxt on EPSCs, in the context of LTD. In addition to the new data, we have substantially modified the text to clarify that Oxt does not potentiate the canonical LTD mechanism itself.